# GROKKING AT THE EDGE OF NUMERICAL STABILITY

**Lucas Prieto, Melih Barsbey, Pedro A.M. Mediano**[*]**, Tolga Birdal**[*]
Department of Computing
Imperial College London

## ABSTRACT

Grokking, or sudden generalization that occurs after prolonged overfitting, is a surprising phenomenon that has challenged our understanding of deep learning. While a lot of progress has been made in understanding grokking, it is still not clear why generalization is delayed and why grokking often does not happen without regularization. In this work we argue that without regularization, grokking tasks push models to the edge of numerical stability, introducing floating point errors in the Softmax that we refer to as *Softmax Collapse* (SC). We show that SC prevents grokking and that mitigating SC leads to grokking *without* regularization. Investigating the root cause of SC, we find that beyond the point of overfitting, the gradients strongly align with what we call the *naïve loss minimization* (NLM) direction. This component of the gradient does not change the predictions of the model but decreases the loss by scaling the logits, usually through the scaling of the weights along their current direction. We show that this scaling of the logits explains the delay in generalization characteristic of grokking, and eventually leads to SC, stopping learning altogether. To validate these hypotheses, we introduce two key contributions that mitigate the issues faced in grokking tasks: (i) $\mathrm{StableMax}$, a new activation function that prevents SC and enables grokking without regularization, and (ii) $\perp\mathrm{Grad}$, a training algorithm that leads to quick generalization in grokking tasks by preventing NLM altogether. These contributions provide new insights into grokking, shedding light on its delayed generalization, reliance on regularization, and the effectiveness of known grokking-inducing methods. Code for this paper can be found at: `https://github.com/LucasPrietoAl/grokking-at-the-edge-of-numerical-stability`.

## 1 INTRODUCTION

Deep learning has been transformative for a variety of fields such as natural language processing (Devlin et al., 2019), computer vision (Krizhevsky et al., 2012), geometry processing (Qi et al., 2017), and 3D vision (Deng et al., 2018). This rapid proliferation has brought with it surprising phenomena that defy the predictions of classical statistical learning theory.

In this paper we explore one such recently observed phenomenon known as *grokking*, first described by Power et al. (2022) as a sudden and unexpected generalization occurring after prolonged overfitting. Although predominantly studied in algorithmic tasks like modular addition or multiplication, recent findings suggest that grokking may be a more pervasive phenomenon, also manifesting in more complex tasks involving vision and language (Lv et al., 2024; Humayun et al., 2024).

Prior research has consistently observed grokking in settings that involve some form of regularization, such as weight decay (Barak et al., 2022; Power et al., 2022; Nanda et al., 2023). This pattern has motivated investigations into the implicit biases introduced by weight decay, suggesting it may be critical to triggering delayed generalization. For instance, Liu et al. (2023a) argued that weight norms need to be in a narrow range or "Goldilocks Zone" for generalization. Similarly, Varma et al. (2023) highlighted weight efficiency of generalizing solutions, and Nanda et al. (2023) argued that weight decay favors simpler, more generalizable solutions. However, recent works have argued that regularization may not be necessary for grokking, at least on shallow networks with Mean Squared

---

[*]Joint senior authors, equal contribution

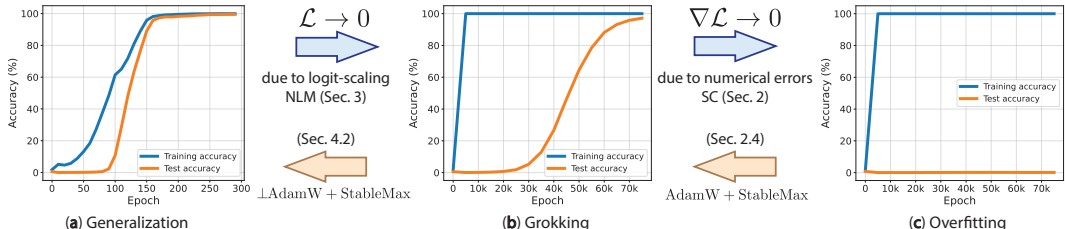

Figure 1: Our contributions demonstrated through results obtained in addition modulo 113 task. We show that the delay in generalization induced by NLM can be reversed using the proposed $\perp$AdamW ((**a**) and (**b**)) and that the numerical errors that lead to overfitting instead of grokking can be avoided by using the proposed StableMax ((**b**) and (**c**)).

Error (MSE) loss (Kumar et al., 2024; Lyu et al., 2024; Gromov, 2023). These works tie grokking to a transition from lazy training (Chizat et al., 2018) to feature learning. Despite this ongoing work, several aspects in this framing of grokking remain unclear. These include why grokking tasks induce lazy training and why weight decay is often needed to enter the feature learning regime when using deeper models or cross-entropy (CE) loss.

Here we propose a novel account of grokking, outlined in Fig. 1, that explains several of the main unanswered questions in the grokking literature. We start by showing that without regularization, grokking is prevented by absorption errors in the Softmax, which we call *Softmax Collapse* (SC). These errors result in zero terms in the gradient and put an end to learning, sometimes before any progress is made in the test performance, resulting in complete overfitting (Fig. 1, **c**). We then argue that SC is caused by what we call *Naïve Loss Minimization* (NLM), as the gradient becomes aligned with a direction that corresponds to scaling up the logits by a constant. While scaling up all the logits does not change the model predictions, it does reduce the CE loss for a network that has reached 100% training accuracy, with the downside that this eventually leads to numerical errors in Softmax. Our findings provide explanations for several key aspects of grokking, including (i) the delayed onset of generalization, (ii) why grokking is often absent without regularization, and (iii) why existing methods designed to induce grokking are effective.

To validate our hypothesis that SC is responsible for the absence of grokking without regularization, we introduce **StableMax** as a more numerically stable replacement to Softmax in CE loss. This simple change takes models from complete overfitting to grokking (Fig. 1, **c** to **b**) *without* regularization, in settings where it is normally not observed without it. Similarly, we validate that NLM is responsible for delaying generalization (Fig. 1, **a** to **b**) and leading to SC by introducing a new optimizer $\perp$Grad, which only preserves the part of the gradient that is orthogonal to the NLM direction. By doing this, $\perp$Grad quickly leads to generalization without the initial overfitting phase that defines grokking (Fig. 1, **b** to **a**).

Our primary contributions are as follows:

- We observe that cases of overfitting without grokking are due to floating point errors caused by extreme values in the Softmax function, which we term Softmax Collapse (SC; Sec. 3).
- We show that interventions to avoid SC, like greater floating point precision or a new, numerically stable version of Softmax (StableMax), cause grokking in settings where it was previously absent without regularization (Sec. 3.3).
- We observe that models move towards SC because overfitting and cross-entropy loss push the model in a direction of uncontrolled logit growth, which we refer to as Naïve Loss Minimization (NLM; Sec. 4).
- We demonstrate that NLM can be avoided through a novel optimizer, $\perp$Grad, which removes the delay in generalization (Sec. 5).

## 2 SETUP

### 2.1 DATASETS

We show our findings on the most commonly studied grokking datasets, outlined in this section.

**I. Modular arithmetic**. The main results in this paper are shown on arithmetic modulo 113 (Power et al., 2022; Nanda et al., 2023). This is a family of supervised learning tasks where two one-hot encoded inputs representing integers $a, b < p$ are used to predict the target $y = a * b \mod p$, where $*$ is some binary operation and $p$ is a prime number. In most of our results, the binary operation is addition, but we show additional results with multiplication and subtraction.

Modular arithmetic tasks are characterized by a binary operation and a dataset size, with different behaviors being observed for different dataset sizes on the same binary operation. In these settings, we describe the dataset sizes as the percentage of the $113^2$ possible pairs that are used for training, with the rest of the data being used for testing as in Nanda et al. (2023) and Power et al. (2022). Our main results use a 40%/60% train/test split but we also include results using 60%/40% and 70%/30%. The input integers are represented as one-hot vectors.

**II. Sparse parity**. We also validate some of our results on the Sparse Parity task outlined in Barak et al. (2022). This is a supervised learning setting where the target is the parity of $k$ bits out of a binary vector of length $n$, with $k \ll n$. In this work we use 2000 samples, split evenly between train and test data and we describe instances of this task by specifying the values of $n$ and $k$.

**III. MNIST**. Finally, we provide some results on a subset the classic image classification dataset MNIST (Deng, 2012). For our experiments, we use a subset of 200 training samples from the training set as in Liu et al. (2023b), with evaluation on the full test set.

## 2.2 Models

We study the grokking phenomenon on these datasets using a 2-hidden layer multi-layer perceptron (MLP) of width 200 as in Liu et al. (2023a) and a one-layer transformer with 4 attention heads as Nanda et al. (2023) and Power et al. (2022). We train both of these models in a full batch setting, using ReLU activations and cross-entropy loss with AdamW and SGD, as well as our own variants of these optimizers, $\perp$AdamW and $\perp$SGD. Unless specified otherwise we set the weight decay parameter $\lambda = 0$. For modular arithmetic datasets, inputs are concatenated as the input of the MLP resulting in a 226 dimensional vector, and treated as separate tokens in the case of the transformer.

## 3 Softmax Collapse: Floating Point Errors Prevent Grokking

Given our current understanding of grokking, it is surprising that it happens without regularization for some dataset sizes, but regularization becomes crucial as dataset size decreases (Power et al., 2022). In this section we highlight that looking at datasets at the boundary of these two regimes reveals that without weight decay, grokking sometimes starts before abruptly stopping (Fig. 2). We show that this is caused by floating point errors in the $\mathrm{Softmax}$ that lead the gradients from a large fraction of the samples to become zero. We refer to this phenomenon as Softmax Collapse.

### 3.1 Softmax Collapse

In modern neural network implementations, Floating Point (FP) arithmetic is ubiquitous for representing and computing parameters, activations, and gradients. While FP numbers enable efficient decimal computations, they introduce numerical inaccuracies. This section focuses on *absorption errors*, as a specific class of FP arithmetic failure. We will use the symbol $\dot{=}$ to refer to equality under FP arithmetic.

**Definition 1** (Absorption Errors). *Let $a, b \in \mathbb{R} \setminus \{0\}$ be floating point numbers in a system with base $\beta$ and $p$ significand bits. Denote their exponents by $e_a$ and $e_b$, respectively. An* absorption error *occurs in the computation of $a + b$ (denoted $a + b \dot{=} a$) if*

$$e_a - e_b \geq p.$$

*In this case, after exponent alignment, the significand of $b$ is shifted right by at least $p$ digits, and $b$ cannot be represented in the available precision, resulting in $a + b \dot{=} a$.*

Intuitively, absorption errors can occur during FP addition when operands have significantly different magnitudes. For $float32$ the base $\beta$ is 2 and $p = 24$ bits, meaning that adding any number smaller than $2^{-(p-1)} = 2^{-23}$ to 1 will leave 1 unchanged. $2^{-23}$ is the machine epsilon for float32.

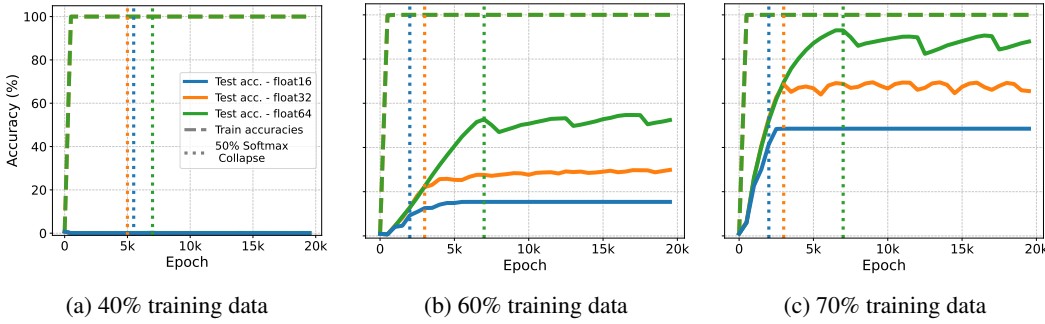

(a) 40% training data          (b) 60% training data          (c) 70% training data

Figure 2: As dataset size increases (subplots **a** to **c**), MLPs trained on modular addition begin to generalize without regularization until this is stopped by SC making the gradient from a large fraction of the samples equal to zero. This stopping point comes earlier for $\mathrm{float}32$ than $\mathrm{float}64$ and with small enough datasets it comes before the model makes any progress on test accuracy.

**Absorption errors in the $\mathrm{Softmax}$**. The $\mathrm{Softmax}$ function is a fundamental component in numerous deep learning architectures, serving as an activation function or a key element in attention mechanisms. In this case, we focus on its application within the Softmax Cross-Entropy (SCE) loss:

**Definition 2** (Softmax Cross-Entropy (SCE) loss). *For a neural network $f$ and a data point $\boldsymbol{x}$ with label $y$, we define $\boldsymbol{z} \coloneqq f(\boldsymbol{x})$ and $z_y$ as the logit corresponding to the true class $y$. We express the SCE loss as well as its equivalent numerically more stable formulation as:*

$$\mathcal{L}_{\mathrm{SCE}}(f(\boldsymbol{x}), y) = -\log\left(\frac{e^{z_y}}{\sum_{k=1}^{n} e^{z_k}}\right) = -z_y + \max(\boldsymbol{z}) + \log\left(\sum_{k=1}^{n} e^{z_k - \max(\boldsymbol{z})}\right) \tag{1}$$

Unfortunately, even the rightmost (comparatively more stable) variant does not address this problem, since the kind of FP errors discussed in this work appear in the sum. While the $\mathrm{Softmax}$ function outputs are bounded between 0 and 1, the intermediate calculations involve summing exponentials of both positive and negative logits. These values can span several orders of magnitude, particularly in scenarios with large logits where the loss approaches zero. This wide range of values creates conditions that lead to absorption errors – leading to the phenomenon we call *Softmax Collapse*.

**Definition 3** (Softmax Collapse (SC)). *A specific case of absorption error occurs when, for a given sample $\boldsymbol{x}$, the logit from the correct class $z_y$ is significantly larger than the logits for all other classes. This floating-point absorption of smaller terms, which we call **Softmax Collapse**, occurs when:*

$$\sum_{k=1}^{n} e^{z_k} \doteq e^{z_y} , \tag{2}$$

*in which case the SCE loss becomes:*

$$\mathcal{L}_{\mathrm{SCE}}(f(\boldsymbol{x}), y) \doteq -\log\left(\frac{e^{z_y}}{e^{z_y}}\right) = 0 . \tag{3}$$

Thus, during SC the loss becomes identical to zero. Furthermore, for the correct class, the gradients become zero as well:

$$\frac{\partial \mathcal{L}_{SCE}}{\partial z_c} = \frac{e^{z_c}}{\sum_{k=1}^{n} e^{z_k}} - \mathbb{1}_{\{c=y\}} \doteq 1 - \mathbb{1}_{\{c=y\}} . \tag{4}$$

While weights that contribute to the wrong classes can still get negative updates, we show that disappearance of the gradients from the correct classes is enough to inhibit grokking (Fig. 2). We validate this in App. B.1 with an explicit intervention, showing that artificially setting the gradients from the correct class to zero stops generalization in a very similar way to what we observe in Fig. 2.

### 3.2 EVIDENCE OF SOFTMAX COLLAPSE IN GROKKING TASKS

Grokking is often studied using dataset sizes for which the delay in generalization is significant, which is usually when the dataset is small but just large enough that generalization is possible. In

this regime, regularization seems necessary for grokking and no improvement in test performance is observed without it (Nanda et al., 2023). However, a fact that has received less attention is that grokking can happen without regularization if the dataset is large enough (Power et al., 2022).

Here we hypothesize that as the size of the dataset decreases, overfitting becomes easier and Softmax Collapse (SC) happens earlier. To quantify this, we train an MLP without regularization on modular addition using different levels of FP precision, and calculate at every training epoch the fraction of samples that result in SC as per Eq. (2). The results support our hypothesis that SC is responsible for the model's failure to generalize (Fig. 2). Specifically, we see that generalization stops when SC begins – and that this happens earlier under $\mathrm{float}32$ than under $\mathrm{float}64$ (Fig. 2b). Furthermore, this point is reached earlier as the dataset size decreases until it is reached before making any progress in the test accuracy, resulting in the common picture of no grokking without regularization (Fig. 2a).

### 3.3 PREVENTING SOFTMAX COLLAPSE LEADS TO GROKKING

To validate the importance of FP errors in stopping grokking, we show that methods to avoid SC lead to generalization on all the common grokking tasks on both MLPs and transformers. We introduce the following methods to postpone the appearance of FP errors.

**Increasing floating point precision**. The simplest way to avoid SC is to extend the FP precision from $\mathrm{float}32$ to $\mathrm{float}64$ for the Softmax calculation. We see in Fig. 2 that networks trained using $\mathrm{float}64$ in the Softmax face SC later in training which allows for a further increase in test performance. Conversely, using $\mathrm{float}16$ leads to SC earlier in training, leading to lower test performance. While this approach works as expected, FP precision cannot be extended indefinitely to allow for generalization as seen in the lack of grokking in Fig. 2a.

**StableMax Cross Entropy (StCE) Loss**. As demonstrated above, SC is caused by adding the exponentials of very large positive and negative logits in the Softmax. To avoid these extreme summands, we propose using a softer version of Softmax to transform logits into probabilities before calculating the CE Loss:

**Definition 4** (StableMax). *We introduce a numerically stable version of the* Softmax *as:*

$$\mathrm{StableMax}(x_i) := \frac{s(x_i)}{\sum_j s(x_j)}, \qquad (5)$$

*where*

$$s(x) := \begin{cases} x + 1 & \text{if } x \geq 0, \\ \frac{1}{1-x} & \text{if } x < 0 \end{cases}. \qquad (6)$$

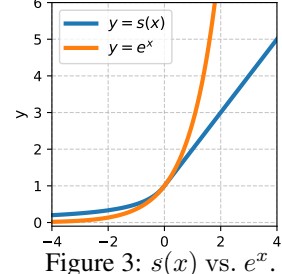

Figure 3: $s(x)$ vs. $e^x$.

As seen in Fig. 3, $s(\cdot)$ is a simple ramp function that scales linearly instead of exponentially when $x \geq 0$ and also approaches 0 more slowly than the exponential function when $x < 0$. This is similar to the Softplus function (Dugas et al., 2000) but approaches 0 more slowly with negative logits, further reducing the risk of absorption errors.

**Proposition 1.** StableMax *is a modified* Softmax, *i.e.* $\mathrm{StableMax}(x_i) = \mathrm{Softmax}(g(x_i))$ *where*

$$g(x) = \begin{cases} \log(x + 1) & \text{if } x \geq 0, \\ -\log(-x + 1) & \text{if } x < 0 \end{cases}. \qquad (7)$$

The proof of this Proposition is presented in App. A. We then define the numerically stable analogue of $\mathcal{L}_{\mathrm{SCE}}$ as $\mathcal{L}_{\mathrm{StCE}}(f(\mathbf{x}), y) = -\log(\mathrm{StableMax}(z_y))$, where $z_y$ again corresponds to the logit of the true class $y$.

To show that StCE indeed addresses the problems posed by SC, we repeat our experiments in Sec. 3.2 by replacing Softmax with StableMax. Our results, presented in Fig. 4, indeed show that StableMax leads to grokking in commonly studied settings *without* regularization. Notably, this happens while the norm of the weights increases substantially (Fig. 4, middle). This suggests that while weight decay may lead to both grokking and a decreasing weight norm, the decreasing

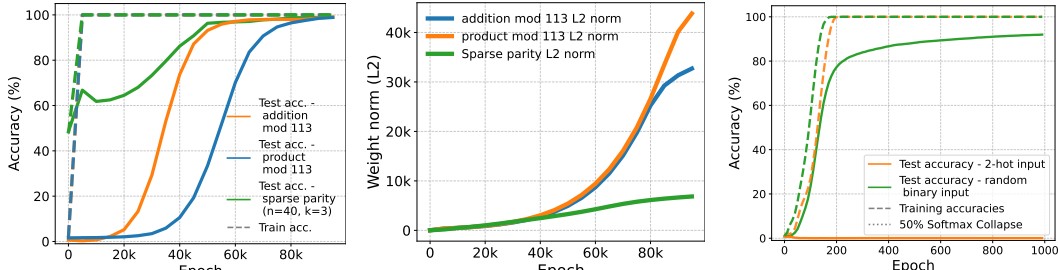

Figure 4: (**left**) Grokking with StCE loss and no regularization on three common grokking datasets using an MLP with 2 hidden layers of width 200. We use 40% of all pairs modulo 113 which is the same setting as Fig. 2a where regular SCE gets stuck at random level performance (random level is 50% for sparse parity). (**middle**) Evolution of model weight norms during training for the same models and tasks. This shows that grokking induced without weight decay does not follow the commonly observed trend of rapidly decreasing weight norm during generalization. (**right**) Changing input representations turns modular addition into regular machine learning tasks with train and test accuracy increasing in tandem, see Sec. 4.

weight norm is not necessary for grokking. Overall, these results i) provide additional evidence for the importance of SC in preventing grokking, ii) suggest a novel activation function to address this problem, and iii) show that regularization or weight norm modification is not *necessary* for grokking.

## 4 DIAGNOSING THE CAUSES OF SOFTMAX COLLAPSE

In the previous section we have shown that FP errors arise due to a combination of low losses and large logits, and shown that when FP errors are mitigated, grokking can be observed in conditions where it previously was not. In this section, we dive deeper and ask why extremely low losses and large logits appear in the first place in grokking tasks. We identify two main causes for this tendency: (i) easiness of overfitting in grokking tasks, and (ii) a training dynamic that sees gradients align with what we call *naïve loss minimization* direction. After diagnosing the causes, the following section will use these insights to develop an optimization algorithm that avoids NLM in the first place.

### 4.1 EASE OF OVERFITTING IN GROKKING TASKS

The first important characteristic of grokking tasks that lead to SC is their ease of overfitting. It has been observed that as grokking datasets get larger, overfitting becomes harder, eventually leading to a regime where train and test performances increase in tandem (Power et al., 2022; Nanda et al., 2023; Varma et al., 2023). It has also been shown that generalization can be delayed in the Sparse Parity task by increasing the amount of noise in the input, which makes overfitting easier (Barak et al., 2022). Here we investigate the opposite effect: that by decreasing the dimensionality of the input the data becomes harder to memorize, removing the delay in generalization.

To do this, we investigate the common grokking task of modular addition, but instead of the high-dimensional one-hot representations of the input integers, we use a more compact binary. More specifically, we assign each integer a distinct random binary vector of dimension 14.

Results confirm our hypothesis, showing that as input representations are decreased in dimension, overfitting is prevented and models generalize without need for regularization (Fig. 4, right). This also shows that modular addition only induces grokking depending on the choice of representation. These findings highlight the importance of understanding the training dynamics beyond the point of overfitting (i.e. point of achieving 100% training accuracy), rather than focusing on the specifics of the modular arithmetic tasks as the key to explaining the delay in generalization.

### 4.2 NAÏVE LOSS MINIMIZATION

We next identify a crucial training dynamic that commonly occurs in grokking tasks as a central cause for increasing logits and SC. We find that after reaching 100% training accuracy, gradient updates are dominated by an update direction we term *naïve loss minimization* (NLM). This direction

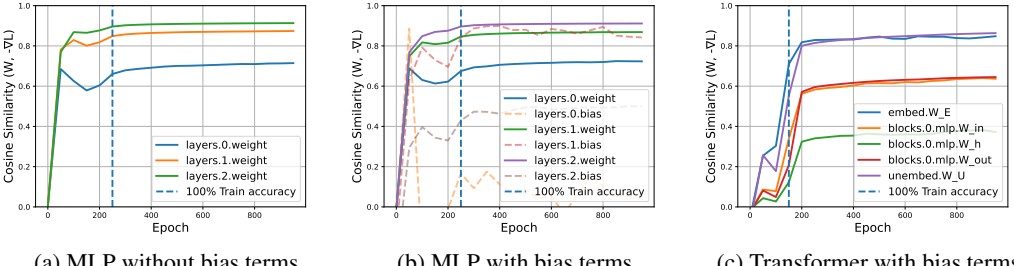

(a) MLP without bias terms     (b) MLP with bias terms     (c) Transformer with bias terms

Figure 5: MLPs with (**a**) and without (**b**) bias terms trained on modular addition receive updates that are significantly aligned with the direction of NLM beyond the point of overfitting. In (**c**) we show these results for a selection of parameters for our one layer transformer. We highlight the embed and unembed matrices as well as the weights of the MLP. These are highlighted in the plot using the notation from Elhage et al. (2021).

does not change the model's decision boundary, but still decreases loss by simply scaling the logits of the predictions, in most cases through scaling of parameters (see below). This means that the logits will continue to increase until they inevitably lead to SC and zero terms in the training gradient. This stops the parameter updates in any direction, including NLM *and* any other useful component that would have been included in the overall gradient. We now define NLM formally, and proceed to discuss why it might commonly be observed to deteriorate training in grokking tasks. Given the input $\mathbf{x} \in \mathcal{X}$, output $y \in \mathcal{Y}$, a predictor $f$ parametrized by $\boldsymbol{\theta} \in \mathbb{R}^m$ that outputs logits $\mathbf{z} = f(\boldsymbol{\theta}; \mathbf{x}) \in \mathbb{R}^{|\mathcal{Y}|}$, and a loss function $\mathcal{L}$, we now define Naïve Loss Minimization.

**Definition 5** (Naïve Loss Minimization (NLM)). *A function $d_{\mathrm{NLM}} : \mathbb{R}^m \to \mathbb{R}^m$ specifies a direction of naïve loss minimization if it decreases the loss,*

$$\mathcal{L}(f(\boldsymbol{\theta} + d_{\mathrm{NLM}}(\boldsymbol{\theta}); \cdot)) < \mathcal{L}(f(\boldsymbol{\theta}; \cdot)), \tag{8}$$

*while satisfying for some $c > 1$:*

$$f(\boldsymbol{\theta} + d_{\mathrm{NLM}}(\boldsymbol{\theta}); \boldsymbol{x}) = cf(\boldsymbol{\theta}; \boldsymbol{x}), \quad \forall \mathbf{x} \in \mathcal{X}, \tag{9}$$

*where $\mathcal{X}$ denotes the input space and $\mathcal{L}(f(\boldsymbol{\theta} + d_{\mathrm{NLM}}(\boldsymbol{\theta}); \cdot))$ is the total loss over the training dataset.*

We find that under a large class of models, namely those that demonstrate *positive homogeneity*, when training beyond 100% training accuracy the direction of the weights is an NLM direction.

**Definition 6** (Positive Homogeneity (Lyu & Li, 2020)). *A function $f$ is positively homogeneous of degree $L > 0$ if for all weights $\boldsymbol{\theta}$, inputs $\mathbf{x}$, and scalars $c > 0$, it satisfies:*

$$f(c\boldsymbol{\theta}; \mathbf{x}) = c^L f(\boldsymbol{\theta}; \mathbf{x}). \tag{10}$$

*When $f$ is a homogeneous neural network, $L$ corresponds to the number of layers.*

In the case of homogeneous networks, training beyond 100% training accuracy, scaling the logits always leads to a decrease in the training loss. Therefore, $d_{\mathrm{NLM}}(\boldsymbol{\theta}) = \alpha\boldsymbol{\theta}$ for $\alpha > 0$ is an NLM direction, as it results in $f(\boldsymbol{\theta} + d_{\mathrm{NLM}}(\boldsymbol{\theta}); \boldsymbol{x}) = f((1 + \alpha)\boldsymbol{\theta}; \boldsymbol{x}) = (1 + \alpha)^L f(\boldsymbol{\theta}; \boldsymbol{x})$, where the second equality follows from Eq. (10).

Many neural network architectures, such as ReLU MLPs and transformers without bias terms, are *positively homogeneous* or *approximately homogeneous* in the case of transformers (Merrill et al., 2020). While more complex deep learning models with skip connections and bias terms are not homogeneous, they have been shown to be quasi-homogeneous (Kunin et al., 2023) and in most cases – including all of the models in this work, the last layer is homogeneous. This means that for non-homogeneous models scaling the weights of the last layer corresponds to a direction of NLM.

The fact that the gradients converge to the direction of the weights has been studied in previous works (Ji & Telgarsky, 2020; 2019; 2018; Lyu & Li, 2020) to prove that homogeneous networks converge in direction under gradient flow and gradient descent (GD), and they perform normalized margin maximization even beyond the point of 100% training accuracy (Lyu & Li, 2020). However,

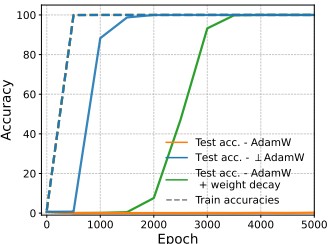 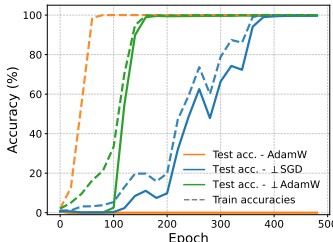 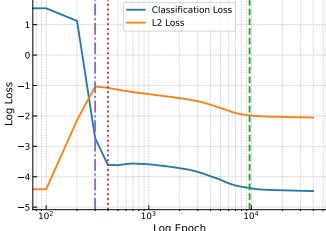

(a) Transformer, subtract. mod 113    (b) MLP, addition mod 113    (c) Trade-off between L2 and SCE

Figure 6: Comparing $\perp$AdamW and $\perp$SGD with baseline optimizers and AdamW with weight decay on (**a**) a transformer trained on subtraction mod 113 and (**b**) an MLP trained on addition modulo 113. In (**c**) we highlight the trade-off between L2 regularization and SCE loss, initially SCE loss is reduced at the cost of increasing the L2 loss but eventually the two losses decrease simultaneously (Sec. 5.2).

we argue that gradient alignment also results in scaling of the logits which can lead to SC and put an end to the margin maximization described in Lyu & Li (2020), when working with limited floating point precision. While we study delayed generalization, the link between training trajectories and generalization is already established in prior art (Birdal et al., 2021; Andreeva et al., 2024).

**Evidence of naïve loss minimization**. In practice, we observe that in MLPs and transformers with and without bias terms, the gradients quickly become aligned with the direction of the weights after the point of overfitting (Fig. 5). Particularly for the later layers of the models, the cosine similarity between the parameter updates and the NLM direction goes up to 0.9 for the output layers. While models with bias terms are not homogeneous and there is no theoretical guarantee that scaling the weights will reduce the SCE loss, in practice, we observe very similar behavior in MLPs with (Fig. 5b) and without (Fig. 5a) bias terms. In the case of a one-layer transformer, the alignment is stronger for the embed and unembed matrices but also substantial for the MLP weights (Fig. 5c).

## 5 MITIGATING NAÏVE LOSS MINIMIZATION LEADS TO GROKKING

While we have shown in Sec. 3 that avoiding numerical instabilities eventually leads to generalization, we can also target the NLM process that causes these numerical issues. To do this, we design an optimizer that only preserves the part of the gradient orthogonal to the direction of the weights.

### 5.1 $\perp$Grad: AN OPTIMIZER TO PREVENT NLM

We propose a new optimizer, $\perp$**Grad** (read "ortho-grad"), that updates the weights based only on the part of the gradient that is orthogonal to the current direction of the weights:

**Definition 7** ($\perp$Grad). *We propose the following update rule for a given iteration $t \in \mathbb{N}$:*

$$\boldsymbol{\theta}_{t+1} = \boldsymbol{\theta}_t - \eta \nabla_\perp \mathcal{L}(\boldsymbol{\theta}_t), \tag{11}$$

*where the orthogonal component of the gradient, $\nabla_\perp \mathcal{L}(\boldsymbol{\theta}_t)$, is obtained by projection onto the hyperplane orthogonal to the current weight vector:*

$$\nabla_\perp \mathcal{L}(\boldsymbol{\theta}_t) = \nabla \mathcal{L}(\boldsymbol{\theta}_t) - \left( \frac{\boldsymbol{\theta}_t^\top \nabla \mathcal{L}(\boldsymbol{\theta}_t)}{\boldsymbol{\theta}_t^\top \boldsymbol{\theta}_t} \right) \boldsymbol{\theta}_t. \tag{12}$$

**Proposition 2.** *Assuming $\nabla_\perp \mathcal{L}(\boldsymbol{\theta}_t) \neq \mathbf{0}$, $\exists \beta > 0$ such that for any learning rate $0 < \eta < \beta$, taking the step $\eta \nabla_\perp \mathcal{L}(\boldsymbol{\theta}_t)$ reduces the loss. In other words, any nonzero $\nabla_\perp \mathcal{L}(\boldsymbol{\theta}_t)$ is a descent direction.*

*Sketch of the proof.* We show that any $\nabla_\perp \mathcal{L}(\boldsymbol{\theta}_t) \in \mathbb{R}^m \backslash \{\mathbf{0}\}$ is a descent direction by demonstrating that $\langle -\nabla_\perp \mathcal{L}(\boldsymbol{\theta}_t), \nabla \mathcal{L}(\boldsymbol{\theta}_t) \rangle < 0$. For a full proof we refer the reader to App. A. ∎

This projection of the gradient can be incorporated into different optimizers. In Fig. 6a, we show results for $\perp$AdamW and $\perp$SGD, the $\perp$Grad versions of AdamW and SGD respectively. These results show that $\perp$Grad optimizers lead to generalization without a phase of initial overfitting, in contexts where no improvement in test performance is usually observed without weight decay. We

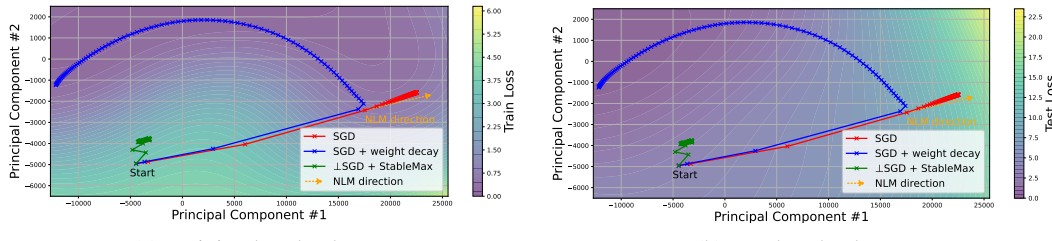

(a) Training loss landscape                  (b) Test loss landscape

Figure 7: Model trajectories in in parameter space projected to 2D over the SCE loss landscape. SGD with weight decay starts along the same trajectory as SGD decreasing the training loss **(a)** but increasing the test loss **(b)**.

note that similar projections of the gradients have been used in other settings to mitigate the effects of momentum in invariant layers (Heo et al., 2021), stabilize training Wang et al. (2024) or as one part in a more complex optimizer (Kosson et al., 2024). We design $\perp \mathrm{Grad}$ as a more precise intervention that directly prevents scaling along the NLM direction.

In Fig. 7, we compare the trajectories of models using SGD with and without weight decay to our new $\perp$SGD optimizer. SGD models start on a similar trajectory, reducing the training loss but increasing the test loss, until the model with weight decay changes direction and starts minimizing both the train and test loss. In contrast, the model using $\perp$SGD moves directly in a direction that minimizes both the train and test loss. While SGD with weight decay eventually reaches a point of lower loss, note that $\perp$SGD reaches 100% test accuracy within 400 iterations (Fig. 6a). Beyond showing how $\perp$SGD prevents NLM, Fig. 7 also suggests that weight decay induces grokking by avoiding NLM. In the following, we highlight that the success of several methods to induce grokking can be explained from this perspective.

## 5.2 EXPLAINING THE SUCCESS OF EXISTING METHODS FOR GROKKING

In light of our findings, we are able to explain the success of several previously proposed methods to induce grokking. We find that these methods also lead to grokking by mitigating NLM and avoiding the FP errors that come with extremely low losses.

**Weight decay**. We have argued that the problem faced in grokking is that the ease of overfitting leads to NLM, which corresponds to scaling up the weights for homogeneous networks. Since weight decay corresponds to pulling back the weights along this same direction at every step during training, it is unsurprising, given our findings, that it is the most reliable way to induce grokking.

To explain why generalization tends to be delayed when using weight decay, as opposed to $\perp \mathrm{Grad}$, we look at it from the perspective of L2 regularization which is equivalent to weight decay for SGD. In Fig. 6c, we see an initial phase where classification loss decreases, at the cost of the L2 loss. Eventually, the decrease in classification loss from NLM stops outweighing the increase in L2 loss, meaning that only updates that are not aligned with the NLM direction are followed. This explains why weight decay leads to generalization in grokking tasks but only after scaling along the NLM direction no longer decreases the overall loss. This balance between weight decay and classification loss is similar to the rotational equilibrium studied in Kosson et al. (2024).

We argue that the main roles of weight decay are preventing floating point errors and preventing NLM. This is in line with recent findings about the role of weight decay in deep learning (D'Angelo et al., 2023) which point to the fact that it increases the effective learning rate and avoids floating point issues when using mixed-precision training in LLMs.

**MSE loss on shallow networks**. While cross-entropy loss can be reduced indefinitely by scaling the logits through NLM, this is not the case with MSE loss. When using MSE loss the logits can overshoot the target, meaning that larger logits often do not lead to a lower MSE loss. This explains why Barak et al. (2022), Kumar et al. (2024), and Lyu et al. (2024) observed grokking with MSE loss without regularization. Interestingly, networks with more than one hidden layer do not generalize in these same settings (Fig. 13).

**Delaying generalization by scaling the weights**. While the lazy training dynamics described in Kumar et al. (2024) explain an important part of why scaling the weights delays generalization,

we show that the reason that regularization is often needed to exit this lazy training regime is that scaling the weights or the logits facilitates SC. In App. D.2, we show that the setting used in Liu et al. (2023b) to induce grokking on MNIST with SCE also induces SC which prevents further learning in the absence of weight decay.

## 6 RELATED WORK

**Grokking**. Power et al. (2022) introduced grokking and showed that weight decay can consistently induce it in algorithmic tasks. Nanda et al. (2023) were able to reverse engineer the inner workings of a grokked transformer and found progress measures for grokking induced by weight decay. Chughtai et al. (2023) generalized the findings from Nanda et al. (2023) and showed grokked networks use group representations to solve group composition tasks, although some of these findings were disputed in Stander et al. (2024) which propose that grokked networks learn a coset based algorithm for these same tasks. Mallinar et al. (2024) has shown that grokking is not specific to neural networks or gradient-based optimization and cannot be predicted from the training or test loss. Varma et al. (2023) argued that grokking is driven by weight decay favoring more efficient solutions and Liu et al. (2023b) hypothesized that the weight norm of the models needs to be in a "Goldilock's zone" to generalize. Kumar et al. (2024) and Lyu et al. (2024) connected grokking to a transition between "lazy training" (Chizat et al., 2018) and feature learning, and Kumar et al. (2024) showed that this can happen without regularization in the case of shallow networks with MSE loss. Grokking has also been described as a phase transition by Žunkovič & Ilievski (2024), Lyu et al. (2024) and Rubin et al. (2024). Humayun et al. (2024) show that in many settings, neural networks undergo grokking-like transitions in their adversarial robustness. This aligns with the findings of Lyu & Li (2020) which attributed this increased robustness to a bias of SGD towards a max-margin solution which was proven for homogeneous models. Beck et al. (2024) also connected grokking to the linear separability of the training data.

**Numerical instability in deep learning**. Numerical instability is a common issue in deep learning Kloberdanz et al. (2022), especially when dealing with mixed precision training D'Angelo et al. (2023). It is known that the $\mathrm{Softmax}$ function is particularly prone to numerical stability problems although this often comes in the form of overflow in the exponential (Kloberdanz et al., 2022) and not from absorption errors in the sum as observed in this case. In the grokking setting, Nanda et al. (2023) showed that the slingshots observed in Thilak et al. (2022) can be explained by a very similar mechanism to the one involved in SC, although Nanda et al. (2023) do not use it to explain any grokking phenomena beyond these spikes that sometimes appear in the training process in grokking tasks. We believe the slingshots observed in Thilak et al. (2022) could be a mechanism to prevent full SC, explaining why slingshots can lead to grokking without weight decay in some settings. This is further discussed in App. H. Issues with numerical instability when training beyond overfitting with increasing learning rates were also observed in Lyu & Li (2020).

## 7 CONCLUSION AND DISCUSSION

In this work, we show that naïve loss minimization (NLM) and floating point errors can explain why generalization is delayed in grokking and why it often does not happen without regularization. Using this insight, we are able to explain the success of existing methods to induce grokking. Motivated by our findings, we further design a simple modification to the $\mathrm{Softmax}$ that induces grokking by avoiding floating point errors and an optimizer that avoids the delay in generalization in grokking by preventing NLM.

**Limitations & future work**. While this work explains several surprising aspects of grokking settings, several questions remain. Notably, we focus our study of NLM on homogeneous or approximately homogeneous models. A a formal characterization quasi-homogenous models could shed light on this kind of dynamics for models including skip connections and bias terms. Additionally, our explanation for why weight decay causes grokking could be enhanced by an analysis of its impact on the effective learning rate as a potential explanation for the sudden nature of grokking.

**Acknowledgments**. This work was supported by the UKRI Centre for Doctoral Training in Safe and Trusted AI [EP/S0233356/1]. TB acknowledges support from the Engineering and Physical Sciences Research Council [grant EP/X011364/1]. TB was supported by a UKRI Future Leaders Fellowship [grant number MR/Y018818/1].

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

APPENDIX

In support of the main paper, App. A presents the proofs for the propositions in the paper, App. B includes additional findings that support our main results, and App. D provides further discussion on conditions that lead to grokking.

## A    PROOFS

*Proof of Prop. 1.*

$$\text{Softmax}\left(g\left(x_i\right)\right) = \frac{e^{g(x_i)}}{\sum_j e^{g(x_j)}} \tag{13}$$

$$= \begin{cases} \frac{e^{\log(x_i+1)}}{\sum_j e^{\log(x_j+1)}} & \text{if } x_i \geq 0, \\ \frac{e^{-\log(-x_i+1)}}{\sum_j e^{-\log(-x_j+1)}} & \text{if } x_i < 0 \end{cases} \tag{14}$$

$$= \begin{cases} \frac{x_i+1}{\sum_j x_j+1} & \text{if } x_i \geq 0, \\ \frac{\frac{1}{-x_i+1}}{\sum_j \frac{1}{-x_j+1}} & \text{if } x_i < 0 \end{cases} \tag{15}$$

$$= \text{StableMax}(x_i). \tag{16}$$

$$\square$$

*Proof of Prop. 2.* To prove that any nonzero $-\nabla_\perp \mathcal{L}(\boldsymbol{\theta}_t)$ is a descent direction, we need to show that $\langle -\nabla_\perp \mathcal{L}(\boldsymbol{\theta}_t), \nabla \mathcal{L}(\boldsymbol{\theta}_t) \rangle < 0$, assuming $\nabla_\perp \mathcal{L}(\boldsymbol{\theta}_t) \neq \boldsymbol{0}$:

$$\left\langle \nabla\mathcal{L}(\boldsymbol{\theta}_t), -\nabla\mathcal{L}(\boldsymbol{\theta}_t) + \left(\frac{\boldsymbol{\theta}_t^\top \nabla\mathcal{L}(\boldsymbol{\theta}_t)}{\boldsymbol{\theta}_t^\top \boldsymbol{\theta}_t}\right)\boldsymbol{\theta}_t \right\rangle \leq 0. \tag{17}$$

Expanding this yields:

$$-\|\nabla\mathcal{L}(\boldsymbol{\theta}_t)\|_2^2 + \left\langle \nabla\mathcal{L}(\boldsymbol{\theta}_t), \boldsymbol{\theta}_t \frac{\boldsymbol{\theta}_t^\top \nabla\mathcal{L}(\boldsymbol{\theta}_t)}{\boldsymbol{\theta}_t^\top \boldsymbol{\theta}_t} \right\rangle \leq 0. \tag{18}$$

Since the inequality is unaffected by the scaling of the left hand side, we can, without loss of generality, assume that the gradients are normalized, leading to:

$$\left\langle \nabla\mathcal{L}(\boldsymbol{\theta}_t), \boldsymbol{\theta}_t \frac{\boldsymbol{\theta}_t^\top \nabla\mathcal{L}(\boldsymbol{\theta}_t)}{\boldsymbol{\theta}_t^\top \boldsymbol{\theta}_t} \right\rangle \leq 1. \tag{19}$$

Since $\boldsymbol{\theta}_t \frac{\boldsymbol{\theta}_t^\top \nabla\mathcal{L}(\boldsymbol{\theta}_t)}{\boldsymbol{\theta}_t^\top \boldsymbol{\theta}_t}$ denotes the projection of the gradient onto the space spanned by the weights, $\langle \cdot, \cdot \rangle$ will measure the acute angle of incidence and hence Eq. (19) holds, with equality iff $\nabla_\perp \mathcal{L}(\boldsymbol{\theta}_t) = \boldsymbol{0}$, which is prevented by assumption. This proves that $-\nabla_\perp \mathcal{L}(\boldsymbol{\theta}_t)$ is a descent direction while being perpendicular to the weights. $\square$

We note that the $\perp$ Grad stops when $\nabla_\perp \mathcal{L}(\boldsymbol{\theta}_t) = \boldsymbol{0}$. If $\nabla\mathcal{L}(\boldsymbol{\theta}_t) \neq \boldsymbol{0}$, this corresponds to the condition where the gradient is in the same direction with the parameter vector. $\nabla_\perp \mathcal{L}(\boldsymbol{\theta}_t) = \boldsymbol{0}$ can also be the case if $\nabla\mathcal{L}(\boldsymbol{\theta}_t) = \boldsymbol{0}$, which corresponds to the loss function being at a local optimum.

## B    ADDITIONAL FINDINGS

### B.1    FURTHER EVIDENCE THAT SC PREVENTS GROKKING

While SC leads the gradient from correctly predicted samples to be zero, it does not do this for the incorrect classes. To validate that setting the gradients from the correct classes to zero is enough to stop learning, we do this artificially for a model that is generalizing and show that learning stops after this intervention. In Fig. 8 we see that the baseline model shown in green generalizes, but this is stopped at epoch 6000 for the model shown in blue, after we perform this intervention.

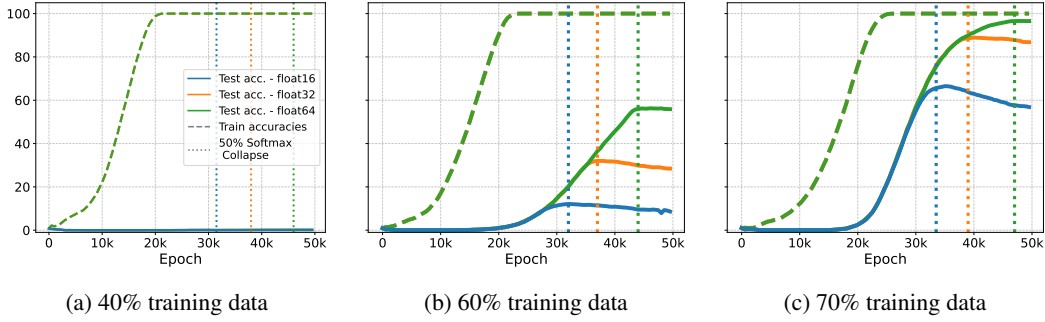

(a) 40% training data      (b) 60% training data      (c) 70% training data

Figure 9: We show that the same dynamics observed in Fig. 2 can be observed with a learning rate scheduler instead of AdamW. This shows that this is not due to an implicit bias of adaptive optimizers.

The intervention is implemented by multiplying the logits for the right classes by 0 at each step after epoch 6000.

## B.2 SGD WITH LEARNING RATE SCHEDULING

To show that our results are not due to the inductive bias of adaptive moments in optimizers like AdamW, we replicate some of the AdamW results using SGD with a learning rate scheduler. Our scheduler is similar to the one in Lyu & Li (2020) except at each step we divide the learning rate by the norm of the full gradient, instead of the loss. In Fig. 9 we observe that SC also puts an end to grokking in this setting.

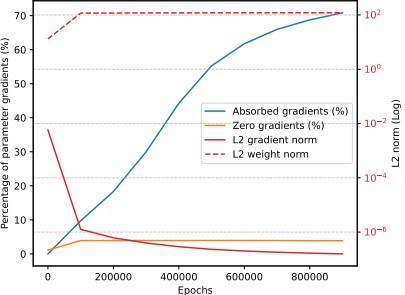

Figure 8: Taking a model that would normally generalize (green) and artificially inducing SC has a very similar effect to the one observed in Fig. 2.

## C EFFECTIVE LEARNING RATE

Unexplored in the main paper, NLM also has the effect of reducing the effective learning rate. For a gradient update using regular gradient descent $\theta_{t+1} = \theta_t - \eta \nabla \mathcal{L}(\theta_t)$ it is easy to see that $||\theta_{t+1} - \theta_t|| \to 0$ as $||\nabla \mathcal{L}(\theta_t)|| \to 0$. This problem has been observed before when training beyond the point of overfitting, for example, Lyu & Li (2020) addressed it by using a loss based learning rate scheduler to keep up with the gradient. Theoretically, an alternative could be to simply extend the duration of training. According to our hypothesis, training for long enough should eventually lead to generalization on grokking tasks if we prevent SC. However, we find that another kind of floating point error can also appears in these settings, namely, gradient absorption errors in the weights.

Figure 10: Gradient absorption errors during training on addition modulo 113.

For a weight $w$, gradient absorption errors happen when a gradient update is small enough that it leaves the weight unchanged. Using the notation outlined in this paper this can be formalized as $w - \eta \frac{\partial \mathcal{L}}{\partial w} \doteq w$. In Fig. 10 we show that this happens for an MLP trained with SGD on modular addition using 30% of the training data. As the norm of the gradient decreases, the percentage of the gradients that are absorbed by the weights increases substantially. Note that the number of gradients that are *exactly* zero remains stable while the number of absorbed gradients increases substantially.

This issue is naturally mitigated by second order moments for adaptive optimizers like Adam and AdamW which is why they do not frequently appear. However, they do prevent us from showing grokking with vanilla gradient descent without any learning rate scheduling.

### C.1 ADDITIONAL WAYS TO INDUCE GROKKING

Beyond the interventions described in the main text, we highlight two additional ways to induce grokking that validate our hypothesis.

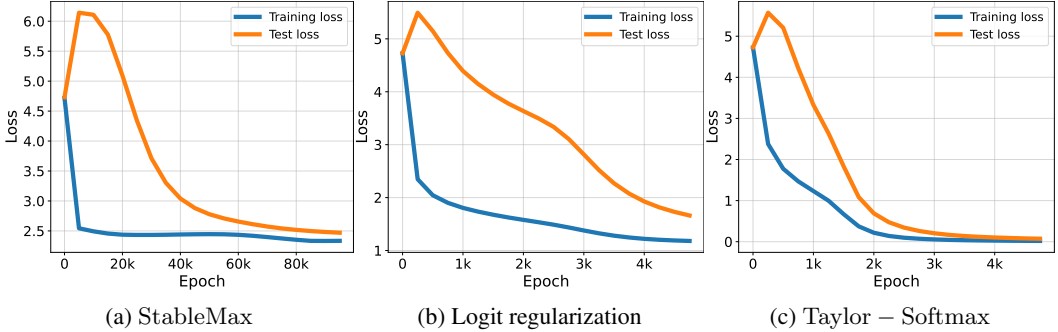

(a) StableMax      (b) Logit regularization      (c) Taylor − Softmax

Figure 11: Train and test losses during grokking induced by three different interventions.

**Logit norm regularization**. Since we argue that uncontrolled scaling of the logits is responsible for delaying grokking and leading to SC, we validate that preventing this scaling of the logits by adding the norm of the logits to the loss, leads to grokking without additional regularization (Fig. 11b).

**Taylor approximation of the Softmax.** We have introduced StableMax as a change to the Softmax that leads to grokking without regularization. The motivation behind this is to prevent values in the sum of the Softmax that are very large or very close to zero. To this end, replacing the exponential with any function that is sub-exponential beyond a certain point should have a similar effect. To demonstrate, we perform a further experiment using the second order Taylor approximation of the exponential

$$e^x \approx 1 + x + \frac{x^2}{2!}, \tag{20}$$

replacing the exp in the Softmax. Since the Taylor approximation is decreasing for $x < 0$, we subtract the minimum logit to avoid this part of the function. We deem this version Taylor − Softmax. In Fig. 11 we see results similar to the ones in Sec. 3.3 but showing the losses instead of the accuracies as well as results for two additional methods to induce grokking. Note that our implementation of Taylor − Softmax (Fig. 11c) introduces an additional implicit regularization simi-

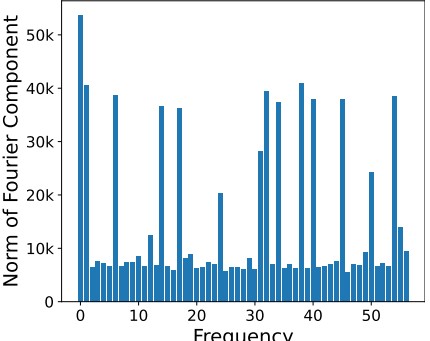

Figure 12: Fourier components of the weights of the output layer of an MLP trained on addition mod 113. Grokking is induced via StableMax and without weight decay.

lar to the one in Fig. 11b, due to the gradient flowing through the subtraction of the mean. While this effectively combines the effects of Fig. 11a and Fig. 11b, leading to grokking faster than the other two methods, our main paper shows results using StableMax as a cleaner intervention that does not introduce this additional regularization effect.

## C.2 SOLUTION LEARNED DURING GROKKING WITHOUT WEIGHT DECAY

Weight decay has been identified as potentially responsible for inducing the periodic structures in the weights studied in Nanda et al. (2023). In Fig. 12 we show that MLPs that grok without weight decay on modular addition show a similar sparsity in Fourier space as the one observed in Nanda et al. (2023). While these are very superficial results, they suggest that these structures can emerge without a weight decay–induced "clean up" phase as described in Nanda et al. (2023).

## D FURTHER DISCUSSION ON CONDITIONS THAT LEAD TO GROKKING

### D.1 L1 REGULARIZATION AND GROKKING

While it has been observed that L1 regularization can lead to grokking in some settings, Nanda et al. (2023) consistently found no grokking with L1 regularization and transformers and this setting has received substantially less attention than weight decay.

We observe that NLM scales the weights along their current direction. This means that larger weights are scaled more than small weights. However, while the sign of the gradient from L1 regularization depends on the sign of the weights, the magnitude of this gradient does not depend on the magnitude of the weights. This means that, particularly on deep networks or transformers with with large weights, L1 can sometimes be insufficient to prevent NLM and the subsequent SC.

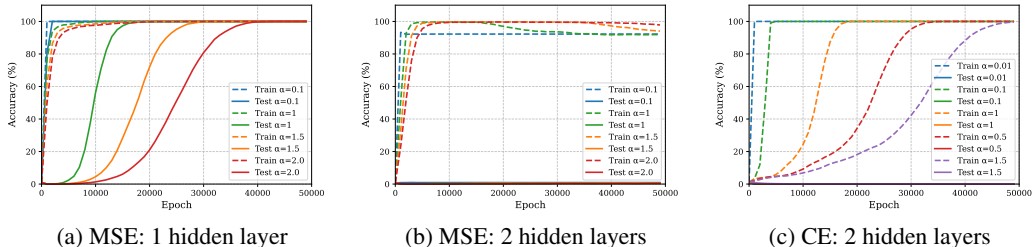

| (a) MSE: 1 hidden layer | (b) MSE: 2 hidden layers | (c) CE: 2 hidden layers |

Figure 13: The $\alpha$ parameter controls generalization in settings where it happens by default. This is the case for shallow networks with MSE loss as shown in subplot (a). However, in deeper networks (b) or networks with CE loss and no regularization (c), $\alpha$ can control the time of over-fitting, but no value of $\alpha$ is enough to trigger grokking.

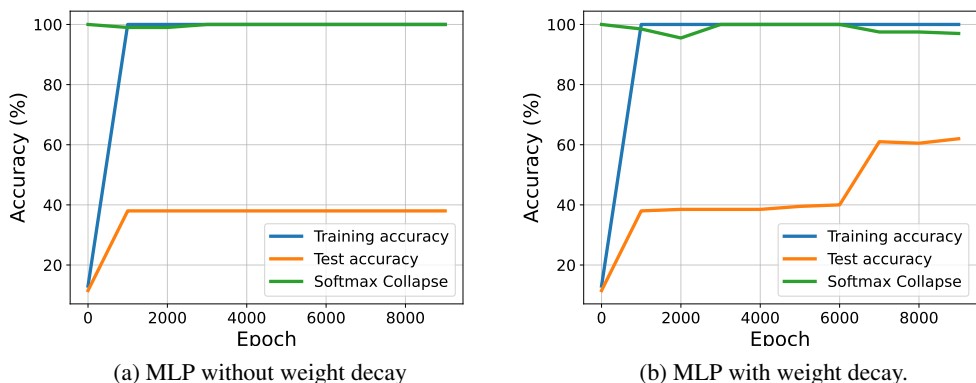

| (a) MLP without weight decay | (b) MLP with weight decay. |

Figure 14: Replicating the grokking on MNIST for weight decay setting from Liu et al. (2023b). We find that MLPs with weights scaled up by 100 operate at the "edge of numerical stability" and in the absence of weight decay, SC eventually reaches 100%, preventing any further generalization. When using weight decay, the weight norm is reduced, mitigating SC and eventually allowing for further generalization as the SC rate drops from 100%.

## D.2 DELAYING GENERALIZATION BY SCALING THE WEIGHTS

**Scaling the logits can delay generalization but not induce it**. Liu et al. (2023a), Kumar et al. (2024) and Lyu et al. (2024) showed that an $\alpha$ parameter multiplying the logits can increase or reduce the delay in generalization. We highlight in Fig. 13 that this is true for cases where generalization happens even without changing the scale of the logits ($\alpha = 1$). However, in most cases when using deeper networks or cross-entropy loss, models do not generalize by default without regularization and we are unable to induce grokking for any value of $\alpha$.

We argue in Sec. 5.2 that the observation in Liu et al. (2023a), Kumar et al. (2024) and Lyu et al. (2024) of grokking without regularization are due to the inductive bias of MSE loss which prevents NLM and leads to grokking in some settings for shallow networks.

**Grokking on MNIST**. We replicate the setting from Liu et al. (2023b) of grokking on MNIST with cross-entropy loss and show that without weight decay, the scaling factor of the weights leads to significant FP errors, preventing grokking from happening until this is alleviated by weight decay.

While SC explains why weight decay is needed to get the jump in performance observed in Fig. 14b. It could also explain why inducing grokking by scaling the weights is less effective when using SCE. While when using MSE loss, Liu et al. (2023a) are able to induce full grokking from random level predictions to close to full training accuracy, the same does not seem to be possible when using SCE. In fact, we see in Fig. 14b that since the beginning of training the rate of SC approaches 100%. This could explain why the observations with cross-entropy loss are not the ones predicted by the lazy training theories outlined in Kumar et al. (2024) which do not take limited floating point precision into account.

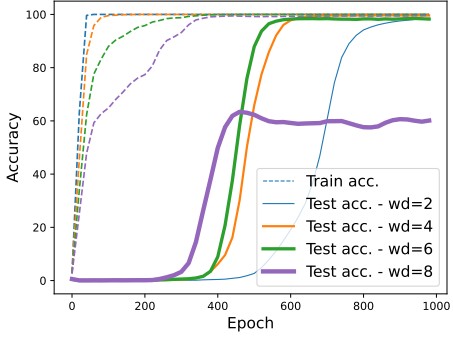
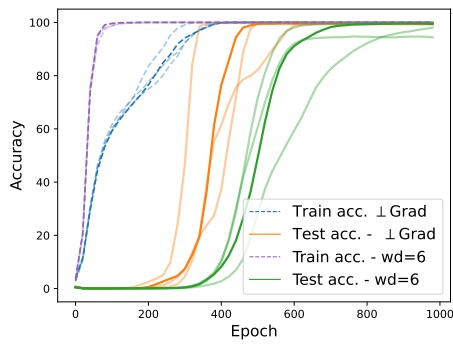

(a) Sweep over values of weight decay      (b) $\perp$Grad vs best performing wd model

Figure 15: Increasing weight decay (WD) for an MLP trained on modular addition with AdamW reduces the delay in generalization up to a point where WD prevents convergence Fig. 15a. Without any tunable hyper-parameters and without WD, $\perp$Grad leads to grokking faster than the best model with WD Fig. 15b.

## E   $\perp$Grad AND WEIGHT DECAY

In Fig. 15, we provide a more in depth comparison of $\perp$Grad and weight decay. Fig. 15a highlights that increasing the weight decay multiplier leads to a smaller delay in generalization, but only up to a point. In this concrete setting, a weight decay multiplier of 8, prevents the model from fully generalizing (Fig. 15a). We then compare the best value of weight decay in this setting to $\perp$Grad, which does not require any hyper-parameter tuning. Fig. 15b shows that $\perp$Grad leads to faster grokking even when compared to a tuned value of weight decay. Note that the models with weight decay overfit immediately before grokking while $\perp$Grad reaches 100% train and test accuracies almost at the same time.

## F   ALTERNATIVES TO StableMax IN PREVENTING SC

While any intervention that prevents SC should lead to grokking or generalization, Fig. 16 shows that scaling the temperature of the Softmax is not enough to prevent SC and label smoothing does prevent SC and lead to some generalization, but at the cost of introducing another inductive bias that prevents full generalization and leads to qualitatively different behavior. By comparison, the simple change introduced in StableMax prevents SC and leads to grokking, serving as a validation for our hypothesis that gradient descent leads to grokking by default, unless this is stopped by SC.

Figure 16: StableMax prevents SC and leads to grokking while temperature scaling with $T = 1e5$ only gradually delays SC, and label smoothing does prevent SC but at the cost of keeping the model from fully generalizing.

## G   StableMax AND $\perp$Grad IN REALISTIC SETTINGS

While StableMax and $\perp$Grad are designed as interventions to show that preventing SC leads to grokking and preventing NLM leads to generalization (Fig. 1), in this section we explore if these methods are applicable in more realistic settings like language modeling with GPT2-small or ResNets trained on image classification. We train GPT2-Small for 1 epoch on WikiText-103 using a batch size of 16, a block size of 512, a learning rate of $5e-4$ and a weight decay of 0.01 using AdamW. The architecture is the regular GPT2-Small architecture from Radford et al. (2019), trained with a cosine schedule and 1000 steps of warm-up.

For CIFAR10, CIFAR100 and Imagenet-1k (Russakovsky et al., 2015), our baseline is a ResNet18 with SCE loss trained with SGD 0.9 momentum and $1e-4$ weight decay. We use standard data transformations such as random crop and random horizontal flip and a step learning rate scheduler every 30 epochs for a full training run of 100 epochs. With respect to this baseline we report results replacing the Softmax with StableMax in the loss function, as well as replacing SGD with $\perp$SGD. Since test labels for Imagenet-1k are not publicly available, we use the validation set as a test set and tune hyper-parameters on a fraction of the training set.

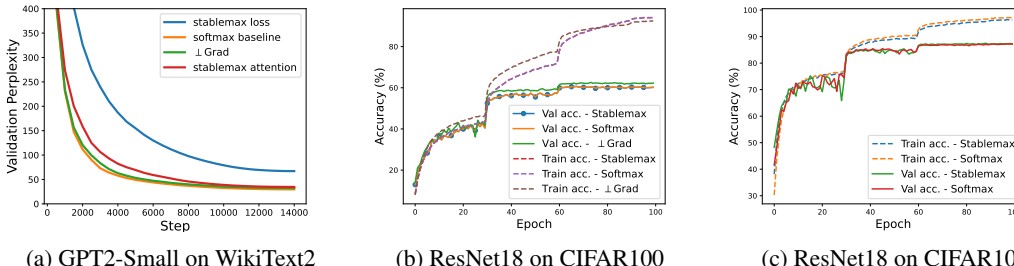

| (a) GPT2-Small on WikiText2 | (b) ResNet18 on CIFAR100 | (c) ResNet18 on CIFAR10 |

Figure 17: Comparing Stablemax and $\perp$Grad to AdamW with SCE on text data Fig. 17a and image data Fig. 17c. For the GPT2-small results in Fig. 17a, we also include the results of replacing the $\mathrm{Softmax}$ in the attention mechanism with $\mathrm{StableMax}$.

| Method | CIFAR10 | CIFAR100 | ImageNet-1k | WikiText-103 (Top-5) |
|---|---|---|---|---|
| Softmax CE | $87.17\% \pm 0.2$ | $59.98\% \pm 0.4$ | $69.33\% \pm 0.04$ | $60.48\% \pm 0.04$ |
| Stablemax CE | $87.01\% \pm 0.2$ | $60.63\% \pm 0.4$ | $65.87\% \pm 0.22$ | $51.85\% \pm 0.47$ |
| $\perp$Grad | $87.22\% \pm 0.2$ | $62.69\% \pm 0.1$ | $68.95\% \pm 0.03$ | $59.64\% \pm 0.04$ |
| Stablemax Attention | – | – | – | $58.52\% \pm 0.04$ |

Table 1: For the methods introduced in this paper, we report accuracies with standard deviations across five seeds for the CIFAR datasets and three seeds for Imagenet-1k and WikiText-103. We report Top-5 accuracy in the case of WikiText-103.

## H   SC AND THE SLINGSHOT EFFECT

Thilak et al. (2022) observed that spikes in the training loss appear when training on grokking tasks with adaptive optimizers like Adam, and that these spikes can lead to generalization without weight decay. Although Nanda et al. (2023) showed that slingshots are not necessary for grokking, it is still unclear what mechanism of adaptive gradient optimizers induces this behavior and why it leads to generalization. In light of the results in this paper, we believe that slingshots could lead to generalization because they prevent full SC. Nanda et al. (2023) pointed out that something like SC could be responsible for these slingshots. One possible mechanism would be that zero gradients for some samples due to SC rapidly diminish the second-order moments leading to a large update or slingshot which moves the model away from full SC, although more research would be needed to properly show this.

While related to our work, slingshots are a different kind of instability which only appears with adaptive optimizers and can allow grokking. In contrast, we identify SC as a very specific issue in the $\mathrm{Softmax}$ that can affect any model trained with SCE, not only the ones trained with adaptive optimizers. Additionally SC prevents grokking whereas slingshots can lead to it. Wether and how slingshots are cause by SC remains an open research question, with some supporting evidence from Nanda et al. (2023) which show that slingshots can disappear when using $float64$.

## I   ADDITIONAL DETAILS ABOUT FLOATING POINTS

Beyond our main results, we found that in some cases, grokking could be stopped before SC due to the $\epsilon$ parameter in Adam being too large. While the $\epsilon$ term is designed to give numerical stability to the gradients, in settings with extremely low losses and gradients, the second order moments can be dominated by the $\epsilon$ term, putting an end to learning where it would have continued with a smaller $\epsilon$ value. This echoes the results in Thilak et al. (2022) which shows that increasing $\epsilon$ halts slingshots and grokking, with Nanda et al. (2023) also alluding to the $\epsilon$ parameter being important in some cases.

Surprisingly, we also found that a simple re-implementation of $torch.nn.functional.log\_softmax$ that does not use the official CUDA kernels can lead the models to keep learning beyond the point where the loss is exactly 0 and some gradients should be 0 with appropriate calculation, outperforming the official implementation for grokking tasks. Learning eventually also stops in this setting and this seems more like a quirk of how gradients are calculated in PyTorch in the absence of an explicitly defined backward pass.

