# OpenReview forum: "Grokking at the Edge of Numerical Stability"
_ICLR.cc/2025/Conference — ICLR 2025 Poster_

### Official Review · Reviewer_2L9F · 2024-11-03

**Soundness:** 3
**Presentation:** 3
**Contribution:** 2
**Rating:** 6
**Confidence:** 3

**Summary:**

This paper addresses the phenomenon of grokking in deep learning, where a model unexpectedly starts to generalize well after a period of severe overfitting. To combat this, the authors propose two primary methods: (1) a numerically stable StableMax activation function aimed at mitigating softmax collapse (SC), and (2) ⊥Grad (orthogonal gradient optimization) to counteract Naïve Loss Minimization (NLM), where gradients become aligned in a way that fails to drive effective learning. These methods are tested on a range of tasks, with a specific focus on scenarios where delayed generalization is a challenge.

**Strengths:**

**Novel Approach to Softmax Collapse**: The introduction of the StableMax activation is an innovative approach to addressing potential issues with Softmax, providing a novel perspective on controlling model output stability.

**Orthogonal Gradients for Effective Training**: The concept of ⊥Grad is compelling and introduces a new approach to optimization that could be useful in a variety of training settings where standard gradient descent shows limitations.

**Comprehensive Analysis**: The paper provides detailed insights into the mechanics of grokking and includes empirical examples to illustrate the phenomenon.

**Weaknesses:**

**Overemphasis on StableMax**: The proposed StableMax activation may be overengineered, as issues with softmax stability are often handled with simpler methods, like temperature scaling or loss smoothing. The authors could provide a comparison with such methods.
The practical necessity and advantage of StableMax might be limited in many standard applications.

**Underdeveloped Exploration of ⊥Grad**: Although ⊥Grad is promising, the current study lacks a thorough exploration of how it compares to more traditional optimization methods across a variety of tasks. Additional benchmarking could help clarify its broader applicability and effectiveness.

**Questions:**

1. How does StableMax compare to standard techniques like loss smoothing? If StableMax’s advantage lies mainly in mitigating SC, could similar results be achieved with regularization techniques already in use? Some experiments with loss smoothing and temperature scaling to establish a fair comparison could be beneficial to the paper.

2. Is ⊥Grad beneficial for tasks where overfitting is not prominent? Understanding if this method generalizes to other contexts could add depth to the findings.

3. ⊥Grad prevents scaling in the logits; is there any impact on margin and generalization, or is it compensated because the norm of weights remains low?
The paper would benefit from additional experiments in varied settings to benchmark ⊥Grad against standard optimizers and even on non grokking tasks.

4. What is the computational overhead of implementing StableMax and ⊥Grad? Practical adoption would require insight into the trade-off between stability and additional computational complexity.

---

> ### Author Response · Authors · 2024-11-22
> **Response to reviewer 2L9F**
>
> We thank the reviewer for the constructive feedback. Below, we address the reviewers questions and concerns. While we appreciate the reviewer's detailed feedback, we want to highlight that we consider the key contributions of this paper to be more about convincingly diagnosing the problems faced in grokking than about the solutions proposed. It is commonly believed in the literature that grokking with cross-entropy loss requires weight decay (or some regularization), and several explanations have been proposed as to why this is the case. In contrast, we show that gradient descent leads to grokking by default (without weight decay), unless this is stopped by floating point errors and explain why these floating point errors appear in grokking (due to NLM). To show this, we introduce StableMax as the simplest change to the Softmax that keeps it from collapse and $\perp$Grad as the most direct intervention to prevent naïve loss minimization (NLM).
>
>
> **Comparing StableMax with label smoothing and temperature scaling:** Thank you for the suggestion. We performed additional experiments to compare label smoothing, temperature scaling and StableMax. We found that even with very large temperature (T=1e+5), models eventually face softmax collapse (SC), and do so very quickly when using optimizers with momentum like Adam. Label smoothing does prevent SC and leads to some generalization, although label smoothing introduces other inductive biases that lead to imperfect generalization and qualitatively different behavior than the one commonly observed in grokking (Figure 16). We believe StableMax provides a targeted intervention, where a single change to the Softmax prevents SC and is the difference between grokking and overfitting. These results have been added to the paper in Figure 16 (Appendix F).
>
> **Evaluating $\perp$Grad in realistic settings:** We have conducted additional experiments comparing $\perp$Grad to other methods and show that it is competitive with our baselines in Imagenet and WikiText, and outperforms them in CIFAR100. However, as suggested by the reviewer the main benefits of this method seem to come in settings wehre overfitting is prominent. These results are included in Appendix G and we also show the full results for $\perp$Grad in the table below.
>
>
> | **Method**             | **CIFAR10**        | **CIFAR100**       | **ImageNet-1k** | **WikiText-103 (Top-5)** |
> |------------------------|--------------------|--------------------|-----------------|--------------------------|
> | Softmax CE             | 87.17\% ± 0.2      | 59.98\% ± 0.4       | 69.76\%          | 59.58\%                  |
> | $\perp$Grad              | 87.22\% ± 0.2      | 62.69\% ± 0.1       | 68.9\%           | 59.53\%                  |
>
> > **Q3:** $\perp$Grad prevents scaling in the logits; is there any impact on margin and generalization, or is it compensated because the norm of weights remains low?
>
> We thank the reviewer for this interesting question. When studying margin maximization in neural networks, the **normalized** margin is commonly studied for the precise reason that margin maximization through scaling all the logits does not lead to generalization [1]. Therefore, the fact that $\perp$Grad prevents scaling in the logits should not (and empirically does not) affect generalization.
>
> > **Q4:** What is the computational overhead of implementing StableMax and $\perp$Grad? Practical adoption would require insight into the trade-off between stability and additional computational complexity.
>
> We thank the reviewer for pointing this out, while StableMax is as fast as Softmax and introduces no additional computational overhead, the gradient projection in $\perp$Grad does account for around 38\% of the training time, in the case of a GPT2-Small trained on WikiText-103 and 34\% for a Resnet18 trained on Imagenet. While we mainly use $\perp$Grad to show to highlight the role of NLM in grokking, if one wants to apply this to more realistic settings, this should be a consideration.
>
> We hope answers and additional experiments have addressed the reviewers concerns, and would allow them to consider updating their score.
>
> [1] Lyu, Kaifeng, and Jian Li. "Gradient descent maximizes the margin of homogeneous neural networks." arXiv preprint arXiv:1906.05890 (2019).

---

> > ### Comment · Reviewer_2L9F · 2024-11-24
> >
> > Thanks to the authors for their detailed clarifications and the additional experiments provided. In consequence, I am upgrading my score.

---

### Official Review · Reviewer_NUmH · 2024-11-04

**Soundness:** 3
**Presentation:** 3
**Contribution:** 3
**Rating:** 8
**Confidence:** 4

**Summary:**

This paper investigates why grokking happens and why it often does not happen without regularization. They show that lack of regularization in tasks where grokking is observed, can lead to overfitting, which is caused by floating point errors in the softmax. Mitigating this softmax collapse (SC) leads to grokking. Further, they show the root cause of SC to be the alignment of gradients with a naive loss minimization (NLM) direction, which does not improve generalization, even though it reduces train loss via scaling. Finally, they propose a StableMax activation that prevents SC and a training algorithm $\perp$ Grad that leads to faster generalization, by amplifying gradients orthogonal to the NLM direction.

**Strengths:**

The paper advances our understanding of why generalization or grokking may not happen without regularization and proposes ways to induce grokking or faster generalization via StableMax and $\perp$ Grad.

The paper is well-written and easy to understand. The claims are well supported by the presented results.

**Weaknesses:**

There are no major weaknesses. Some minor concerns are as follows.

- The phase ‘grokking tasks’ can be reworded as ‘tasks where grokking is commonly observed’, since there is no formal definition of grokking tasks.
- The readability can be improved by better placement of the figures (same page as the corresponding discussion).
- In lines 302-304, the digits 1- can be replaces with i) for better readability.

**Questions:**

- How do other factors such as initialization scale, learning rate, etc. interact with grokking?
- Do the results shown in Fig. 4 correspond to one the settings shown in Fig. 2?

---

> ### Author Response · Authors · 2024-11-22
> **Response to reviewer  NUmH**
>
> We thank the reviewer for this positive feedback. We have made some changes to address the minor concerns.
>
> > **Q1:** How do other factors such as initialization scale, learning rate, etc. interact with grokking?
>
> Our observations align with the literature in observing that learning rate can speed up grokking up to a point and that scaling the weights can delay grokking. We additionally observe in Appendix D.2 that scaling the weights also scales the logits, leading to faster Softmax Collapse and explaining how delayed generalization can be induced in MNIST by scaling the weights and using cross-entropy loss.
>
> > **Q2:** Do the results shown in Fig. 4 correspond to one the settings shown in Fig. 2
>
> Yes, thank you for pointing this out, we have updated the caption fro Fig 4 to make this clearer.

---

> > ### Comment · Reviewer_NUmH · 2024-11-22
> >
> > Thank you for the clarifications, I will maintain my score.

---

### Official Review · Reviewer_yRJ9 · 2024-11-04

**Soundness:** 2
**Presentation:** 3
**Contribution:** 2
**Rating:** 8
**Confidence:** 4

**Summary:**

The paper studies grokking. It found that when the loss function is cross-entropy loss, the floating point errors caused by softmax prevents the model from generalization if trained without any regularization. They proposed a modified loss function and a modified optimizer that  enable the model to generalize without regularization.

**Strengths:**

This work studies grokking from an interesting view and provided an explanation for why the model is trapped in an overfitting regime.

**Weaknesses:**

- The application of the proposed method is limited. It only turns from overfitting without generalization to generalization when there is no regularization and the original loss is cross-entropy loss. Is ⊥AdamW still better than AdamW with non-zero weight decay? E.g. can you compare ⊥AdamW against AdamW with various non-zero weight decay values for transformers and MLP on modular addition task?
- The proposed method is only tested on some toy models such as two-layer neural networks and one-layer transformer. It would be more convincing to apply the method on deeper transformers, e.g. the same architecture as GPT2-small.
- Sec 4.1 compares one-hot encoding with binary and scalar representation to show overfitting is harder, which is not fair:
    - 1. the one-hot vector does not have any information of the integers, but binary and scalar vectors do contain some info.
    - 2. the model trained on binary and scalar vectors has fewer parameters, making overfitting harder.
A fair comparison would be using randomly generated d-dim vectors on a uniform sphere as the low-dim representation, and using a model with larger width to make sure the number of params is roughly the same.

**Questions:**

- What’s the difference between absorption error and Roundoff error?

- Why setting the gradient of correctly predicted (overfitted) samples to zero prevents generalization? Is there an intuitive explanation?

- In Figure 1c, can grokking still occur after careful hyper parameters(initialization scale, learning rate) tunning?

- [1] found that one-layer transformer trained with float32 precision has loss spikes and use float64 for more stable training. This implies that the slingshot mechanism [2] may be caused by numeric stability. Can softmax collapse explain slingshot?

[1] Progress measures for grokking via mechanistic interpretability

[2] The Slingshot Mechanism: An Empirical Study of Adaptive Optimizers and the Grokking Phenomenon

---

> ### Author Response · Authors · 2024-11-22
> **Response to reviewer yRJ9**
>
> We thank the reviewer for the constructive feedback. We provide answers to the reviewers questions below:
>
> **Comparing $\perp$AdamW to AdamW with weight decay:** Thank you for the suggestion. In addition to the results with weight decay we have provided in Figure 6, we have added a comparison of $\perp$AdamW against AdamW with the weight decay parameter selected to induce grokking as early as possible. The results show that weight decay can anticipate grokking up to a point, beyond which weight decay prevents generalization. Even without any tunable hyper-parameters, $\perp$Grad leads to faster grokking than any value of weight decay. These results have been included in Appendix E.
>
> **Evaluation in more realistic settings:** We agree with the reviewer that it is interesting to observe the behavior of our interventions in the beyond-the-toy settings. We take this suggestion into account and added experiments with more realistic models like Resnet18 and GPT2-Small on datasets like Imagenet and GPT2-Small. These results are included and discussed under **Evaluation in more realistic settings** in the general response and have been added to the paper in Appendix G.
>
>
> **One hot vs binary encoding:** We thank the reviewer for the insightful comment. We agree that the previous setup was not a fair comparison as the binary and scalar representations introduce information about the structure of the integers making the task easier as well as making overfitting harder. As a remedy, we have updated this setup such that the inputs involve random permutations  applied to the integers before converting them into binary representations and concatenating them, thereby removing any information about the integers being encoded. The results indicate that reducing the dimension of the input data leads to direct generalization, even if this new representation does not introduce any additional information about the integers and we adjust the number of parameters (Increasing the hidden size from 200 to 280  for a roughly equal parameter count when decreasing the input size from 226 to 14). These results have been updated in Figure 4.
>
> **Absorption vs roundoff errors:** A Roundoff error is any difference between the result of a sum with floating point numbers and the true result of that sum. Absorption is a special case of Roundoff error that happens when adding a very large and a very small floating point number such that the smaller number is "absorbed".
>
> Q2: Why setting the gradient of correctly predicted (overfitted) samples to zero prevents generalization? Is there an intuitive explanation?
>
> **Zero-gradients of overfitted samples:** While the training accuracy is 100\%, cross-entropy loss should never be 0 in the absence of floating point errors. It has been proven in [3] that homogeneous models perform normalized margin maximization beyond the point of 100\% training accuracy and we believe this is the process that leads to grokking unless it is stopped by Softmax Collapse (SC).
>
> Q3: In Figure 1c, can grokking still occur after careful hyper parameters(initialization scale, learning rate) tuning?
>
> **Grokking and hyper-parameter tuning:**
> We do not believe that grokking can be induced in this setting without introducing an intervention that prevents SC. In Figure 1c we show that grokking does not occur even after training for 80,000 epochs in a setting where generalization happens in less than 200 epochs with $\perp$Grad. Both of these use AdamW with regular initialization scale and the same very common hyper-parameters (lr=0.001 and default AdamW with weight decay set to 0). Additionally, Figure 2a shows that in this same setting, the gradients coming from the correct class for most of the samples become 0, preventing generalization beyond that point.
>
> Q4: [1] found that one-layer transformer trained with float32 precision has loss spikes and use float64 for more stable training. This implies that the slingshot mechanism [2] may be caused by numeric stability. Can softmax collapse explain slingshot?
>
> **Can our findings explain slingshot mechanism?**
> We thank the reviewer for bringing this interesting insight. Indeed, based on the observations from [1], we believe softmax collapse plays an important role in the creation of slingshots, when this is combined with the second order moments in Adam (which were found to be necessary for slignshots in [2]). We hypothesize that zero gradients caused by SC lead to very small second order moments resulting in slingshots when these weights get non-zero gradients, but we do not have enough experiments on this front to be sure. This is briefly mentioned under `numerical instability in deep learning' in Related Work.

---

> > ### Comment · Reviewer_yRJ9 · 2024-12-02
> >
> > Thank you for your response. I will raise my score.

---

### Official Review · Reviewer_sm8G · 2024-11-06

**Soundness:** 2
**Presentation:** 2
**Contribution:** 2
**Rating:** 5
**Confidence:** 4

**Summary:**

The paper investigates the phenomenon of grokking, where deep learning models suddenly generalize after a period of overfitting, and it proposes new mechanisms that explain why generalization is often delayed. The authors identify Softmax Collapse, a numerical instability in Softmax, as a key factor preventing grokking in models trained without Weight Decay. To address this, they introduce "StableMax," a modified Softmax function that reduces numerical errors, and "⊥Grad," an optimizer that excludes the naive loss minimization direction.  The latter refers to an optimization path that reduces the loss by scaling logits without changing model predictions and potentially leading to numerical errors.

**Strengths:**

- The paper is well-written and easy to follow.
- The research problem is conceptually well-motivated and aligns with current challenges in understanding grokking.
- The hypothesis presented, that Softmax Collapse inhibits grokking, is interesting.

**Weaknesses:**

1. **Unclear Justification for Proposed  Adjustments Over Established Regularization Techniques** While the paper suggests several adjustments to address generalization delay, it remains unclear if and why these modifications should be preferred over established techniques such as weight decay, particularly at larger scales. A comparative analysis with standard approaches would strengthen the argument for these adjustments.

2. **StableMax Analysis:** The introduction of StableMax is a substantial change to the training paradigm. However, the paper lacks an in-depth analysis of its properties, such as the impact on the optimization landscape and the types of solutions this modified model converges towards. Given the importance of StableMax, its introduction should be supported by either extensive experimental validation or a rigorous theoretical justification.

3. **Optimizer in Section 5.1:** Similar to StableMax, the optimizer proposed in Section 5.1 would benefit from additional empirical or theoretical analysis. Specifically, how it compares with other optimizers in terms of convergence behavior and stability beyond the context of grokking tasks.

4. **Insufficient Evidence for Naive Loss Minimization:** The evidence provided for the claim of naive loss minimization is limited. Existing literature, such as that referenced work of Lyu and Li (2019)​[1] (in Appendix K.1 ), shows that even after achieving 100% training accuracy, gradient descent with cross-entropy loss implicitly performs margin maximization. This result seems to contradict the authors’ hypothesis that the optimization follows a direction of naive loss minimization, where the function remains unchanged across the entire input space (as described in Equation 9).  Further evidence is needed to validate naive loss minimization from this established understanding of implicit bias in gradient descent.

**Questions:**

1. Could the authors further discuss naive loss minimization in relation to the max-margin implicit bias as described in [1]? In particular the claim that the direction of the optimization trajectory is the one that does not change the prediction for all data points in the input space (Equation 9).


2. Could the authors clarify why the proposed adjustments should be prioritized over traditional methods like weight decay?


3. Can the authors elaborate on how the proposed optimizer relates to [2]?


4. Is there a connection between the projected optimizer in this paper and the rotational equilibrium update discussed in [3] as an effect of weight decay?


5. For Figure 2, could the authors specify the task on which the models are trained?


6. In Figure 6, do the authors have experiments demonstrating the impact of different weight decay values on the delay in generalization?

[1] Lyu, Kaifeng, and Jian Li. "Gradient descent maximizes the margin of homogeneous neural networks." _arXiv preprint arXiv:1906.05890_ (2019).


[2] Heo, Byeongho, et al. "Adamp: Slowing down the slowdown for momentum optimizers on scale-invariant weights." _arXiv preprint arXiv:2006.08217_ (2020).


[3] Kosson, Atli, Bettina Messmer, and Martin Jaggi. "Rotational equilibrium: How weight decay balances learning across neural networks." _arXiv preprint arXiv:2305.17212_ (2023).

---

> ### Author Response · Authors · 2024-11-22
> **Response to reviewer sm8G**
>
> We thank the reviewer for the constructive feedback and provide detailed answers to their questions below.
>
>
> **Connection to Lyu and Li (2019):** The connection to the work from Lyu and Li ([1]) is very relevant. While we define the NLM direction as not changing the predictions for any data points (Equation 9), we argue that in practice, models strongly "align" with the NLM direction with large cosine similarities above 0.9 in the MLP case (Figure 5), but they do not exactly follow the NLM direction. The part of the gradient that is orthogonal to the NLM direction does increase the margin as predicted by Lyu and Li and validated by the success of $\perp$Grad which only preserves this part of the gradient. Lyu and Li also run into numerical issues in prolonged training beyond 100\% accuracy (Appendix L.2). In their case these issues can be fixed by using float64, but we show that in cases of grokking this is not enough, and diagnose the particular kind of floating point issues to be Softmax Collapse. To take the connection to this paper further, we believe the max-margin implicit bias described in Lyu and Li (2019) is the reason we should observe grokking by default. The reason this is not observed in the literature when using cross-entropy loss is that Softmax Collapse stops this margin maximization process. While Lyu and Li prove this max-margin property for gradient flow and gradient descent, the proofs do not account for limited floating point precision. We have updated the paper to make it clearer that our findings are in line with this established implicit bias of GD (Section 4.2).
>
> **Comparison to weight decay** StableMax and $\perp$Grad are mainly presented as targeted interventions to validate our hypothesis that floating point errors caused by NLM inhibit grokking and are not intended to replace weight decay. However, we have added results showing that weight decay only anticipates grokking up to a point (weight decay=6 in our case) beyond which it prevents generalization (Figure 15a). We compare an MLP trained on modular addition using the tuned weight decay value and show that $\perp$Grad still leads to faster grokking than weight decay without requiring tuning of the weight decay parameter (Appendix E, Figure 15b).
>
> **Additional literature:** We thank the reviewer for pointing out this relevant literature. The rotational equilibrium described in [3] could explain why weight decay behaves like $\perp$Grad, but only after a delay. In the terms of [3], the delayed generalization in grokking could correspond to the \emph{transient phase}, with generalization starting when the model reaches \emph{rotational equilibrium}. More work would be needed to make this connection rigorous. While the optimizer in [2] is designed for scale invariant weights and does not prevent scaling for regular weights, some of the insights about how to handle momentum in these settings could be incorporated to improve the performance of $\perp$Grad when combined with Adam ($\perp$Adam). We have updated Section 5.1 to mention this and cite these works.
>
> Question: For Figure 2, could the authors specify the task on which the models are trained?
>
> Answer: We thank the review for pointing this out, we have updated the figure caption to mention that these results are from modular addition.
>
> **General note:** While this paper's main goal is to gain insight into the grokking, we agree that an exploration of the broader applicability of these methods can strengthen the paper. As such, we have conducted experiments (results in the table under the **General Response**) using GPT2-Small, and Resnet18 that show that our methods behave similarly to our baseline in the context of language modeling (WikiText-103) and image classification (Imagenet-1k, CIFAR100, CIFAR10), with the exception of StableMax which behaves almost identically to softmax when predicting across a limited number of classes (CIFAR100) but decreases performance when predicting over 50,000 tokens (WikiText-103). Fot a more detailed discussion of these new results we point the reviewer to the general response. These results are also included and discussed in the updated paper under Appendix G.

---

> > ### Author Response · Authors · 2024-11-25
> >
> > We would like to thank the reviewer once again for their helpful feedback. We would greatly value the opportunity to engage further in a constructive discussion and remain available to address any additional questions or concerns during the discussion period. In light of the clarifications, additional evaluations, and extended exposition we have provided,  we kindly encourage the reviewer to revisit their assessment and consider adjusting their final rating before the discussion period concludes.

---

> > > ### Comment · Reviewer_sm8G · 2024-12-03
> > >
> > > Thank you for the clarifications and additional experiments, which improved my understanding of StableMax and ⊥Grad. The insights into grokking are interesting, and I appreciate the updated results showing broader applicability.
> > >
> > > However, I still believe further analysis of the optimization trajectory, particularly its relationship to grokking and generalization, is necessary to fully capture the phenomenon’s dynamics. I have increased my score to reflect the improved manuscript while encouraging additional exploration in this area. Thank you for your thoughtful revisions.

---

> > > > ### Author Response · Authors · 2024-12-03
> > > >
> > > > We thank the reviewer for considering our feedback. We certainly agree that optimization trajectories carry essential information for understanding intruiging properties of deep networks. Indeed, numerous studies have shown key connections between training trajectories and generalization error. Exploring training dynamics, also in the context of grokking is undeniably a promising avenue for future research.
> > > >
> > > > However, this paper is focused on a different but equally critical aspect. Specifically, we identify two pivotal factors--floating-point precision and logit scaling--that, when properly addressed, lead to grokking. We further design targeted interventions to empirically validate our findings. Our mechanistic interpretation not only offers fresh insights into the phenomenon but also sheds light on why certain regularization techniques have been effective.
> > > >
> > > > We believe that, in this specific context, our work makes a significant and unique contribution. While the reviewer's suggested direction is valuable, we feel the need to emphasise that our work addresses a different research direction and question. At no point have we suggested that our framework “fully captures the phenomenon’s dynamics”. Instead, we have focused on making concrete contributions that advance our understanding towards a better theory of grokking.
> > > >
> > > > Given the unanimously positive ratings from other reviewers and the clear merits of our contributions, we respectfully request the reviewer to reconsider their judgment one last time.

---

### Official Review · Reviewer_Mcra · 2024-11-07

**Soundness:** 3
**Presentation:** 3
**Contribution:** 3
**Rating:** 8
**Confidence:** 4

**Summary:**

This paper uncovers the critical role of numerical stability in grokking, demonstrating that Softmax Collapse  (SC) prevents grokking and that mitigating SC leads to grokking without regularization.

**Strengths:**

- The authors make a novel discovery regarding the crucial role of numerical stability in grokking and identify the Softmax Collapse (SC) phenomenon.
- They reveal that SC results from naive loss minimization (NLM) of cross-entropy loss.
- Based on these theoretical insights, the authors propose two methods—StableMax and $\perp$Grad—to mitigate issues faced in grokking tasks.

**Weaknesses:**

- **Softmax Collapse (SC).**
  - The current definitions in Section 3.1 is somewhat abstract and could be more practically grounded. For example, when using float16, what is the $\epsilon$ in Definition 1, and what specific difference between the two logits in Definition 3 would lead to SC?
  - In transformers, softmax is applied in all self-attention layers, not just the last layer. Does SC also occur in these layers?

- **StableMax.**
  - I am concerned that StableMax may not be suitable for the softmax in attention layers of transformers. Many experiments show that attention matrices in transformers are highly sparse, but the *slower growth rate* of StableMax (compared to softmax) could hinder its ability to model such sparsity.

- **Naive Loss Minimization (NLM).**
  - It is worth noting that this emperical observation has been proven in related works, such as in linear models [1], deep linear networks [2], and homogeneous networks [3]. As training progresses, the "angular velocity" becomes much slower than the growth of the norm, and the directional convergence holds: $\left<-\frac{\nabla L(\theta_t)}{\|\|\nabla L(\theta_t)\|\|},\frac{\theta_t}{\|\|\theta_t\|\|}\right>\to1$.
  - It seems that this explanation holds only for cross-entropy loss. For squared loss, NLM does not appear to hold.

- **$\perp$Grad.**
  - Based on my understanding, the motivation behind $\perp$Grad is to amplify the angular velocity of GD, similar to the effects of normalization layers and weight decay. Additionally, related work [4] also explores how to amplify the angular velocity of GD.
  - Adaptive algorithms like Adam may not encounter the same NLM problem as GD late in training. Would it be effective to use $\perp$Grad+Adam to accelarate the grokking process of Adam?


References:

[1] Ji and Telgarsky. Risk and parameter convergence of logistic regression. COLT 2019.

[2] Ji and Telgarsky. Gradient descent aligns the layers of deep linear networks. ICLR 2019.

[3] Ji and Telgarsky. Directional convergence and alignment in deep learning. NeurIPS 2020.

[4] Wang et al. Achieving Margin Maximization Exponentially Fast via Progressive Norm Rescaling. ICML 2024.

**Questions:**

Refer to Weaknesses.

---

> ### Author Response · Authors · 2024-11-22
> **Response to reviewer Mcra**
>
> We thank the reviewer for the constructive feedback. Below, we address the reviewers questions and suggestions.
>
> **Clearer definition of absorption errors:** We thank the reviewer for pointing this out. Type float32 uses 24 significand bits, meaning that adding any number smaller than $2^{-24}$ to 1 ($1=2^0$) will leave 1 unchanged. $2^{-24}$ is called the machine epsilon for float32. We have now clarified this in the main paper (Section 3.1).
>
> **StableMax in the attention mechanims of a transformer:** Although StableMax is designed as a minimalist intervention to validate our hypothesis that preventing SC leads to grokking, we share the interest of the reviewer in seeing if this could be included in the attention mechanism of a transformer. We have added experiments using the GPT2-Small architecture which show that using StableMax instead of Softmax in the attention mechanism has a small negative impact on performance, from 59.58\% to 58.37\% top-5 accuracy. This suggests that the sparsity induced by softmax may be beneficial in the attention mechanims, as suggested by the reviewer, although the effect is small and more extensive experimentation is needed to conclude this definitively. These results have been added to Appendix G.
>
> **Additional related work:** We agree that NLM is closely related to the phenomena discussed in these works ([1], [2], [3]) as well as [5], and have included a short discussion of this as well as the relevant citations in Section 4.2 . While the convergence of the gradients to the direction of the weights is not a new observation, we highlight that this leads to the scaling of the logits, which can become a problem when we take limited floating point precision into account.
>
> **Relevance for MSE loss:** The reviewer is correct in highlighting that the phenomena we observe do not apply to MSE loss. As discussed in Section 5.2, this actually explains why grokking without regularization has been observed with MSE loss but not with cross-entropy, which was a previously unexplained phenomenon.
>
> **Concerns about $\perp$Grad:** We agree with the reviewer that $\perp$Grad does amplify the angular velocity and is related to methods like [4], which uses progressive rescaling to address a limitation of normalized gradient descent and actually use a projection similar to $\perp$Grad as a way to stabilize their method. We have updated the paper to mention this in Section 5.1 . Regarding the reviewers question about adaptive algorithms, Adam does seem to face the same NLM and SC problem as SGD. As shown in Figure 1, AdamW without weight decay does not generalize and faces SC (Figure 2a), in settings where preventing SC does lead to grokking. Figure 1a also shows that $\perp$Grad+Adam (termed $\perp$AdamW in the paper) does accelerate grokking, as suggested by the reviewer.
>
> [5] Lyu, Kaifeng, and Jian Li. "Gradient descent maximizes the margin of homogeneous neural networks." arXiv preprint arXiv:1906.05890 (2019).

---

> > ### Comment · Reviewer_Mcra · 2024-11-25
> > **good luck**
> >
> > Many thanks to the authors for the detailed response. I feel that this paper provides great insights into grokking caused by numerical stability/instability, and I have raised my score to 8.

---

### Author Response · Authors · 2024-11-22
**General Response**

We thank all reviewers for their constructive feedback. We are happy to see that the reviewers appreciate the **critical role** [Mcra] that the dynamics uncovered in this paper play in grokking and believe that it **advances our understanding** [NUmH] of this phenomenon, with other reviewers agreeing that the work is **interesting** and **well motivated** [sm8G, yRj9] and the proposed solutions **innovative** and **compelling** [2L9F]. We now address the general concerns.

**Evaluation in more realistic settings:** While the StableMax and $\perp$Grad are mainly proposed as interventions to validate our novel hypotheses and advance our understanding of grokking and generalization, we appreciate the reviewers' interest in seeing how these methods would perform in more realistic settings. Accordingly, we have added experiments using ResNet18 on Imagenet-1k, CIFAR100 and CIFAR10, as well as GPT2-Small on Wikitext-103,the results in the table below, show that outside of the extreme overfitting setting they were designed for, $\perp$Grad and StableMax cross-entropy achieve performances which are comparable with our baseline for language modeling and image classification, with $\perp$Grad outperforming our baseline for CIFAR100. StableMax is designed to be as similar as possible to softmax while preventing softmax collapse, therefore, it seems to behave very similarly to softmax in most cases, however, we notice that when StableMax is applied over a large number of classes or tokens (50,000 for GPT2) it does seem to degrade performance from 59.58\% to 52.14\% top-5 test accuracy and a less significant decrease is also observed for Imagenet-1k (69.76\% to 65.76\%). We hypothesize that the difference between softmax and StableMax becomes more significant when applied over large dimensions, where the probabilities assigned to the large number of incorrect classes does not decrease as quickly with StableMax, interfering with the signal from the correct label. In these settings increasing the temperature of StableMax or other interventions might be needed to use StableMax over very large dimensions. Overall, we find that \textit{i}) in cases where numerical instability is a significant risk, StableMax and $\perp$Grad bring substantial benefits from avoiding such instability; and \textit{ii}) when there is no numerical instability, in most cases StableMax and $\perp$Grad have no negative effect (except StableMax applied over large dimensions) -- as expected from our theoretical motivation. The table below is included in Appendix G of the revised paper, and reports test accuracies which are averaged across 5 seeds and show the standard deviation for CIFAR10 and CIFAR100. We are currently running different seeds for Imagenet-1k and WikiText-103 to be included in the final version.


| **Method**             |    **CIFAR10**        |    **CIFAR100**       |       **ImageNet-1k**.   |    **WikiText-103 (Top-5)** |
|------------------------|--------------------|--------------------|-----------------|--------------------------|
| Softmax CE             | 87.17\% ± 0.2      | 59.98\% ± 0.4       |     69.76\%          |    59.58\%                  |
| StableMax CE           | 87.01\% ± 0.2      | 60.63\% ± 0.4       |     65.76\%          |   52.14\%                  |
| $\perp$Grad              | 87.22\% ± 0.2      | 62.69\% ± 0.1       |    68.9\%           |    59.53\%                  |
| StableMax Attention| --                | --                 | --              |    58.37\%                  |

**$\perp$Grad and weight decay:** Following the helpful suggestion from the reviewers, we have added a more detaield comparison of weight decay and $\perp$Grad. Our results show that higher weight decay can anticipate grokking, but only up to a point, beyond which it prevents generalization. We then compare a tuned value of weight decay, chosen to induce grokking as early as possible and see that it is still marginally slower than $\perp$Grad which does not require any hyper-parameters. Additionally, we observe that AdamW with weigh decay overfits very quickly, leaving a larger delay until generalization, while with $\perp$Grad, train and test accuracies reach 100\% almost at the same time. These results have been added to the paper in Figure 15 (Appendix E).


**Related literature:**
We also thank the reviewers for pointing out some relevant literature, particularly, existing works related to $\perp$Grad and NLM. We have updated the paper to include a short discussion of these methods and their differences from $\perp$Grad. As suggested by reviewer 2L9F, we have also included a comparison of StableMax to label smoothing and temperature scaling in the Appendix.

More detailed descriptions of these changes are provided in the answers to individual reviewers.

---

> ### Author Response · Authors · 2024-11-28
>
> We would like to inform the reviewers that, as discussed in our previous response, we have now updated the results table (Table 1 in the paper) to report the mean accuracy and standard deviation across different seeds for Imagenet-1k and WikiText-103. Below is the updated table:
>
> | **Method**            | **CIFAR10**          | **CIFAR100**        | **ImageNet-1k**       | **WikiText-103** (Top-5)     |
> |------------------------|----------------------|---------------------|-----------------------|------------------------------|
> | Softmax CE            | 87.17% ± 0.2        | 59.98% ± 0.4        | 69.33% ± 0.04         | 60.48% ± 0.04               |
> | Stablemax CE          | 87.01% ± 0.2        | 60.63% ± 0.4        | 65.87% ± 0.22         | 51.85% ± 0.47               |
> | $\perp$Grad               | 87.22% ± 0.2        | 62.69% ± 0.1        | 68.95% ± 0.03         | 59.64% ± 0.04               |
> | Stablemax Attention   | --                   | --                  | --                    | 58.52% ± 0.04               |
>
> *For the methods introduced in this paper, we report test accuracies with standard deviations across five seeds for the CIFAR datasets and three seeds for ImageNet-1k and WikiText-103. We report Top-5 accuracy in the case of WikiText-103.*

---

### Public Comment · ~Vimal_Thilak1 · 2024-12-02
**Models used in paper**

I read the paper with great interest and think that this paper is good. I am an author that contributed to [Slingshot Effect]((https://openreview.net/forum?id=OZbn8ULouY) work that is referred to by the authors in this paper. I am curious to know whether the authors used a 2 layer Transformer with the datasets and setup described by Power et al. in their original grokking paper? Looking at lines 519-524, I am curious to understand whether the authors considered some of the observations made by us while training Transformers without with Adam (and no weight decay)? One attempt to resolve instabilities during optimization was to increase epsilon but this resulted in no generalization (and consequently no grokking) which is undesirable.

We also have a discussion on scaling weights and activations of the last layer in an appendix (C.2) in [our paper](https://openreview.net/forum?id=OZbn8ULouY). If the authors find our observations relevant then would they perhaps consider discussing these specific observations as related work in a future revision?

Best,

Vimal

---

### Meta-Review · Area_Chair_ZDra · 2024-12-16

**Metareview:**

This paper aims to explain why grokking appears and disappears (without regularization) for several existing methods to induce grokking. It identifies the crucial role of numerical stability in grokking and identify the Softmax Collapse (SC) phenomenon. Accordingly, the authors develop a simple modification of the Softmax to avoid floating point errors, which will induce grokking. The extensive experiments can support this claim.

It's true that the training dynamics is missing, particularly its relationship to grokking and generalization, which indeed decreases the significance of this work. But anyway, this paper is good in general, and I recommend to accept.

**Additional Comments On Reviewer Discussion:**

All of the reviewers engage with the authors and the discussion clarified several points the reviewers concerned.

---

### Decision · Program_Chairs · 2025-01-22

Accept (Poster)